# Hydrogen sulfide blocks HIV rebound by maintaining mitochondrial bioenergetics and redox homeostasis

Virender Kumar Pal[1,2], Ragini Agrawal[1,2], Srabanti Rakshit[2], Pooja Shekar[3], Diwakar Tumkur Narasimha Murthy[4], Annapurna Vyakarnam[2,5], Amit Singh[1,2]*

[1]Department of Microbiology and Cell Biology, Indian Institute of Science, Bangalore, India; [2]Centre for Infectious Disease Research (CIDR), Indian Institute of Science, Bangalore, India; [3]Bangalore Medical College and Research Institute, Bangalore, India; [4]Department of Internal Medicine, Bangalore Medical College and Research Institute, Bangalore, India; [5]Peter Gorer Department of Immunobiology, School of Immunology and Microbial Sciences, Faculty of Life Sciences & Medicine, Guy's Hospital, King's College London, London, United Kingdom

**Abstract** A fundamental challenge in human immunodeficiency virus (HIV) eradication is to understand how the virus establishes latency, maintains stable cellular reservoirs, and promotes rebound upon interruption of antiretroviral therapy (ART). Here, we discovered an unexpected role of the ubiquitous gasotransmitter hydrogen sulfide ($H_2S$) in HIV latency and reactivation. We show that reactivation of HIV is associated with downregulation of the key $H_2S$ producing enzyme cystathionine-γ-lyase (CTH) and reduction in endogenous $H_2S$. Genetic silencing of CTH disrupts redox homeostasis, impairs mitochondrial function, and remodels the transcriptome of latent cells to trigger HIV reactivation. Chemical complementation of CTH activity using a slow-releasing $H_2S$ donor, GYY4137, suppressed HIV reactivation and diminished virus replication. Mechanistically, GYY4137 blocked HIV reactivation by inducing the Keap1-Nrf2 pathway, inhibiting NF-κB, and recruiting the epigenetic silencer, YY1, to the HIV promoter. In latently infected CD4+ T cells from ART-suppressed human subjects, GYY4137 in combination with ART prevented viral rebound and improved mitochondrial bioenergetics. Moreover, prolonged exposure to GYY4137 exhibited no adverse influence on proviral content or CD4+ T cell subsets, indicating that diminished viral rebound is due to a loss of transcription rather than a selective loss of infected cells. In summary, this work provides mechanistic insight into $H_2S$-mediated suppression of viral rebound and suggests exploration of $H_2S$ donors to maintain HIV in a latent form.

*For correspondence:
asingh@iisc.ac.in

**Competing interest:** The authors declare that no competing interests exist.

## Editor's evaluation

Pal and colleagues show that hydrogen sulfide ($H_2S$) inhibits HIV replication and reactivation by a variety of mechanisms including inhibition of NF-κB and recruitment of the epigenetic silencer, YY1, to the HIV promoter. They further report that $H_2S$ helps in maintaining mitochondrial bioenergetics and redox homeostasis. Altogether, the study provides novel insights into the mechanisms underlying HIV transcription and reactivation.

## Introduction

Human immunodeficiency virus (HIV), the causative agent of acquired immunodeficiency syndrome (AIDS), is responsible for 0.6 million deaths and 1.7 million new infections in 2019 (https://www.who.

int/data/gho/data/themes/hiv-aids). Despite advances in antiretroviral therapy (ART), the persistence of latent but replication-competent HIV in cellular reservoirs is a major barrier to cure (*Finzi et al., 1997*; *Sengupta and Siliciano, 2018*; *Cohn et al., 2020*). Understanding host factors and signaling pathways underlying HIV latency and rebound upon cessation of ART is of the highest importance in the search for an HIV cure (*Deeks et al., 2016*; *Collins et al., 2020*). Host-generated gaseous signaling molecules such as nitric oxide (NO), carbon monoxide (CO), and hydrogen sulfide ($H_2S$) modulate anti-microbial activity, inflammatory response, and metabolism of immune cells to influence bacterial and viral infections (*Chinta et al., 2016*; *Pal et al., 2018*; *Tinajero-Trejo et al., 2013*; *Rahman et al., 2020*; *Shiloh et al., 2008*; *Braverman and Stanley, 2017*). Surprisingly, while circumstantial evidence links NO and CO with HIV-1 infection (*Blond et al., 2000*; *Devadas and Dhawan, 2006*; *Liu et al., 2016*), the role of $H_2S$ remained completely unexplored.

$H_2S$ is primarily synthesized in mammals, including humans, via the reverse transsulfuration pathway using methionine and cysteine metabolizing enzymes cystathionine-gamma-lyase (CSE/CTH), cystathionine-beta-synthase (CBS), and cysteine-aminotransferase (CAT)–3-mercaptopyruvate sulfurtransferase (MPST), respectively (*Pal et al., 2018*). Because $H_2S$ is lipophilic, it rapidly permeates through biological membranes without membrane channels (lipid bilayer permeability, $P_M \geq 0.5 \pm 0.4$ cm/s), dissociates into $HS^-$ and $S^{2-}$ in aqueous solution, and maintains an $HS^-$:$H_2S$ ratio of 3:1 at physiological pH (*Pal et al., 2018*). Interestingly, expression and enzymatic activity of CTH was diminished in human $CD8^+$ T cells and hepatic tissues derived from AIDS patients, respectively (*Hyrcza et al., 2007*; *Martin et al., 2001*). Additionally, the levels of $H_2S$ precursors L-cysteine and methionine were severely limited in AIDS patients (*Hortin et al., 1994*). Also, HIV infection reduces the major cellular antioxidant glutathione (GSH) (*Staal et al., 1992*), whose production is dependent on L-cysteine and $H_2S$ (*Kimura et al., 2010*; *McBean, 2012*). Consequently, supplementation with N-acetyl cysteine (NAC), which is an exogenous source of L-cysteine, has been proposed for AIDS treatment (*Roederer et al., 1993*). Since functional CBS and CTH enzymes are known to promote T cell activation and proliferation (*Miller et al., 2012*), endogenous levels of $H_2S$ can potentiate T cell-based immunity against HIV. Therefore, we think that the endogenous levels and biochemical activity of $H_2S$, which critically depends upon the cysteine and transsulfuration pathway (*Mathai et al., 2009*), are likely to be important determinants of HIV infection.

Calibrated production of $H_2S$ improves the functioning of various immune cells by protecting them from deleterious oxidants, stimulating mitochondrial oxidative phosphorylation (OXPHOS), and counteracting systemic inflammation (*Pal et al., 2018*; *Modis et al., 2014*; *Fu et al., 2012*). In contrast, supraphysiological concentrations of $H_2S$ exert cytotoxicity by inhibiting OXPHOS, promoting inflammation, and inducing redox imbalance (*Pal et al., 2018*; *Modis et al., 2014*; *Fu et al., 2012*). These $H_2S$-specific physiological changes are also crucial for HIV-1 infection. First, chronic inflammation contributes to HIV-1 pathogenesis by affecting innate and adaptive immune responses (*Deeks et al., 2013*), impairing recovery of $CD4^+$ T cells post-ART (*Nakanjako et al., 2011*), increasing the risk of comorbidities (*Leng and Margolick, 2015*; *Alzahrani et al., 2019*), and worsening organ function (*Tattevin et al., 2007*). Second, HIV-1 preferentially infects $CD4^+$ T cells with high OXPHOS (*Valle-Casuso et al., 2019*), relies on glycolysis for continuous replication (*Hegedus et al., 2014*), and reactivates in response to altered mitochondrial bioenergetics (*Tyagi et al., 2020*). Third, a higher capacity to resist oxidative stress promotes virus latency, whereas increased oxidative stress levels induce reactivation (*Bhaskar et al., 2015*; *Savarino et al., 2009*; *Shytaj et al., 2020*). Fourth, $H_2S$ directly (via S-linked sulfhydration) or indirectly modulates the activity of transcription factors NF-κB and AP-1 (*Sen et al., 2012*; *Xie et al., 2016*) that are well known to trigger HIV-1 reactivation from latency (*Williams et al., 2007*; *Brooks et al., 2003*; *Duverger et al., 2013*). Finally, the thioredoxin/thioredoxin reductase (Trx/TrxR) system crucial for maintaining HIV-1 reservoirs (*Chirullo et al., 2013*) has been shown to regulate $H_2S$-mediated S-persulfidation (*Dóka et al., 2016*). Altogether, many physiological processes vital for HIV-1 pathogenesis overlap with the underlying mechanism of $H_2S$-mediated signaling.

Based on the role of $H_2S$ in regulating redox balance, mitochondrial bioenergetics, and inflammation, we hypothesize that intracellular levels of $H_2S$ modulate the HIV-1 latency and reactivation program. To test this hypothesis, we utilized cell line-based models of HIV-1 latency and $CD4^+$ T cells of HIV-1 infected patients on suppressive ART, which harbors the latent but replication-competent virus. Biochemical and genetic approaches were exploited to investigate the link between $H_2S$ and

HIV-1 latency. Finally, we used NanoString gene expression technology, oxidant-specific fluorescent dyes, and real-time extracellular flux analysis to examine the role of $H_2S$ in mediating HIV-1 persistence by regulating gene expression, redox signaling, and mitochondrial bioenergetics.

## Results

### Diminished biogenesis of endogenous $H_2S$ during HIV-1 reactivation

To investigate the link between HIV-1 latency and $H_2S$, we measured changes in the expression of genes encoding $H_2S$-generating enzymes CBS, CTH, and MPST in monocytic (U1) and lymphocytic (J1.1 and J-Lat) models of HIV-1 latency (*Figure 1A*). The U1 and J1.1/J-Lat cell lines are well-studied models of post-integration latency and were derived from chronically infected clones of a promonocytic (U937) and a T-lymphoid (Jurkat) cell line, respectively (*Folks et al., 1987*; *Perez et al., 1991*; *Jordan et al., 2003*). Both U1 and J1.1/J-Lat show very low basal expression of the HIV-1 genome, which can be induced by several latency reversal agents (LRAs) such as PMA, TNF-α, and prostratin (*Figure 1B*; *Folks et al., 1987*; *Perez et al., 1991*; *Jordan et al., 2003*). First, we confirmed virus reactivation by measuring HIV-1 *gag* transcript in U1 with a low concentration of PMA (5 ng/ml). Treatment of PMA induces detectable expression of *gag* at 12 hr, which continued to increase for the entire 36 hr duration of the experiment (*Figure 1C*). Next, we assessed the mRNA and protein levels of CBS, CTH, and MPST in U1 during virus latency (untreated) and reactivation (PMA-treated). The mRNA and protein expression levels of CTH showed a significant reduction at 24 and 36 hr post-PMA treatment as compared to the untreated control, whereas the expression of CBS was not detected and MPST remained unaffected (*Figure 1D–E*). As an uninfected control, we measured the expression of CBS, CTH, and MPST in U937 cells. In contrast to U1, PMA treatment stimulated the mRNA and protein levels of CTH in U937 (*Figure 1F–G*), while CBS was barely detectable. Also, we directly measured endogenous $H_2S$ levels using a fluorescent probe 7-azido-4-methylcoumarin (AzMc) that quantitatively detects $H_2S$ in living cells (*Chen et al., 2013*). Consistent with the expression data, reactivation of HIV-1 by PMA reduced $H_2S$ generation in U1, whereas $H_2S$ levels were significantly increased by PMA in the uninfected U937 (*Figure 1H–I*).

Similar to U1 monocytic cells, PMA treatment reactivated HIV-1 and reduced the expression of CTH in J1.1 lymphocytic cells but not in the uninfected Jurkat cells in a time-dependent manner (*Figure 1—figure supplement 1A-B*). The expression of CBS was reduced in both J1.1 and Jurkat upon PMA treatment, indicating that only CTH specifically downregulates in response to HIV-1 reactivation (*Figure 1—figure supplement 1A-B*). In J-Lat cells, HIV-1 genome encodes GFP, which facilitates precise measurement of PMA-mediated HIV-1 reactivation from latency by flow cytometry. Treatment with PMA reactivated HIV-1 in a time-dependent manner and diminished the expression of CBS, CTH, and MPST in J-Lat as compared to Jurkat cells (*Figure 1—figure supplement 1C-D*). Taken together, these data indicate that HIV-1 reactivation is associated with diminished biogenesis of endogenous $H_2S$.

### CTH-mediated reactivation of HIV-1 from latency

Our results suggest that $H_2S$ biogenesis via CTH is associated with HIV-1 latency. Therefore, we next asked whether CTH-derived $H_2S$ regulates the reactivation of HIV-1 from latency. To test this idea, we depleted endogenous CTH levels in U1 using RNA interference (RNAi). The short hairpin RNA specific for CTH (U1-shCTH) silenced the expression of CTH mRNA and protein by 90% as compared to non-targeting shRNA (U1-shNT) (*Figure 2A–B* and *Supplementary file 1a*). Moreover, using AzMC probe, we confirmed the reduction in endogenous $H_2S$ levels in U1-shCTH as compared to U1-shNT (*Figure 2—figure supplement 1A-B*). In addition to $H_2S$ production, CTH catalyzes the last step in the reverse transsulfuration pathway to generate cysteine. However, levels of cysteine remain comparable in U1-shCTH and U1-shNT (*Figure 2—figure supplement 1C*), indicating that other routes for cysteine biosynthesis (e.g., de novo and assimilatory pathways) compensate for CTH depletion in U1. Next, we investigated the effect of $H_2S$ depletion via CTH suppression on HIV-1 latency by measuring viral transcription (*gag* transcript), translation (HIV-1 p24 capsid protein), and release (HIV-1 p24 abundance in the cell supernatant). We found that the low basal expression of HIV-1 *gag* in U1-shCTH was stimulated by 16-fold upon depletion of CTH (p=0.0018) (*Figure 2C*). Furthermore, while PMA induced expression of *gag* transcript by 73-fold in U1-shNT, a further enhancement to 200-fold was

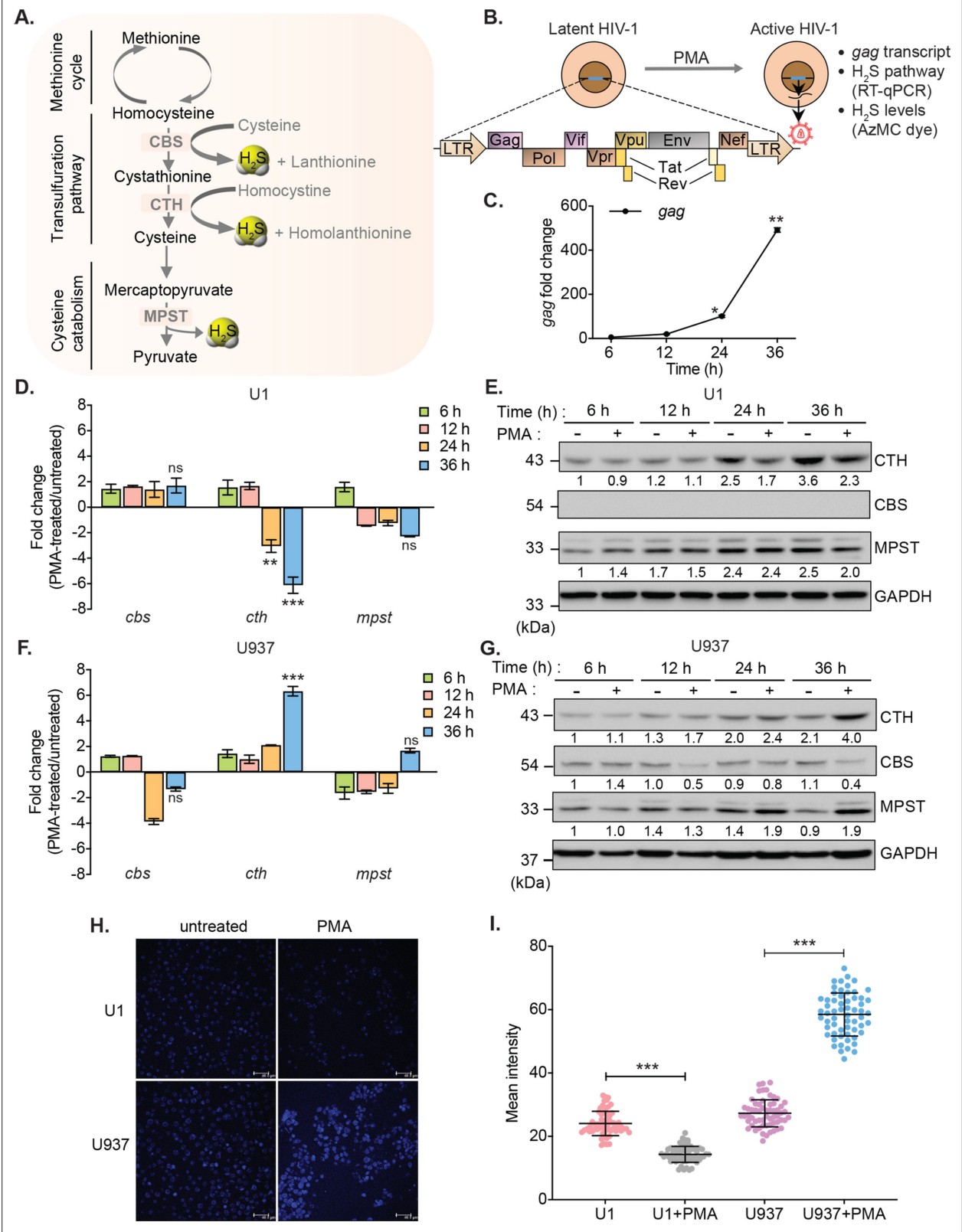

**Figure 1.** HIV-1 reactivation diminishes expression of H$_2$S metabolic enzymes and endogenous H$_2$S levels. (**A**) Schematic showing H$_2$S producing enzymes in mammalian cells. (**B**) Experimental strategy for measuring HIV-1 reactivation and H$_2$S production in U1. (**C**) U1 cells were stimulated with 5 ng/ml PMA and the expression of *gag* transcript was measured at indicated time points. (**D, E**) Time-dependent changes in the expression of CTH, CBS, and MPST at mRNA and protein level during HIV-1 latency (−PMA) and reactivation (+PMA [5 ng/ml]) in U1 cells. (**F, G**) Time-dependent

*Figure 1 continued*

changes in the expression of CTH, CBS, and MPST at mRNA and protein level in U937 cells with or without PMA treatment. Results were quantified by densitometric analysis for CTH, CBS, and MPST band intensities and normalized to GAPDH, using ImageJ software. (**H, I**) U1 and U937 cells were treated with 5 ng/ml PMA for 24 hr or left untreated, stained with AzMC for 30 min at 37°C, and images were acquired using Leica TCS SP5 confocal microscope (**H**). Scale bar represents 40 µm. Average fluorescence intensity was quantified by ImageJ software (**I**). Results are expressed as mean ± standard deviation and are representative of data from three independent experiments. *, p<0.05; **, p<0.01; ***, p<0.001, by two-way ANOVA with Tukey's multiple comparison test.

The online version of this article includes the following figure supplement(s) for figure 1:

**Source data 1.** This file contains the source data used to generate *Figure 1E*.

**Source data 2.** This file contains the source data used to generate *Figure 1G*.

**Source data 3.** This file contains the source data used to make the graphs presented in *Figure 1*.

**Figure supplement 1.** HIV-1 reactivation decreases levels of $H_2S$ metabolizing enzymes in T cell line models of latency.

**Figure supplement 1—source data 1.** This file contains the source data used to generate *Figure 1—figure supplement 1A*.

**Figure supplement 1—source data 2.** This file contains the source data used to generate *Figure 1—figure supplement 1B*.

observed in U1-shCTH (*Figure 2C*). Consistent with this, levels of p24 capsid protein inside cells or released in the supernatant were significantly elevated in U1-shCTH as compared to U1-shNT with (p=0.028) or without PMA-treatment (p=0.015) (*Figure 2D–E*). Deprivation of cysteine in the culture medium does not stimulate viral transcription in U1 (*Figure 2—figure supplement 1D*), indicating that reduction in $H_2S$ levels is likely responsible for the reactivation of HIV-1 in U1-shCTH.

We also depleted CTH levels in J1.1 cells using RNAi and monitored HIV-1 reactivation by assessing intracellular p24 levels under basal conditions. The depletion of CTH triggered HIV-1 reactivation from latency as evident from a sevenfold increase in p24 levels as compared to shNT (*Figure 2—figure supplement 1E*). A minor increase in p24 levels (3.8-fold) was also apparent upon the depletion of CBS in J1.1 (*Figure 2—figure supplement 1E*). Finally, and consistent with U1 and J1.1, genetic silencing of CTH using a CTH specific shRNA or chemical inhibition by propargylglycine (PAG) reactivated HIV-1 in J-Lat cells (*Figure 2—figure supplement 1F-I*). These data indicate that CTH and likely CTH-derived $H_2S$ supports latency and impede the reactivation of HIV-1.

## Altered expression of genes involved in HIV-1 reactivation upon CTH depletion

The above results indicate that CTH is a target controlling HIV-1 reactivation from latency, prompting us to investigate the mechanism. Several pathways that induce HIV-1 reactivation from latency are also influenced by $H_2S$. These include redox signaling (*Banerjee, 2011*), NF-κB pathway (*Gao et al., 2012*), inflammatory response (*Bhatia, 2012*), and mitochondrial bioenergetics (*Szabo et al., 2014*). Hence, we conducted a focused expression profiling of 185 human genes intrinsically linked to HIV-1 reactivation using the NanoString nCounter Technology (see *Supplementary file 1b* for the gene list) to measure absolute amounts of multiple transcripts without reverse transcription. Because depletion of CTH promoted reactivation of the HIV-1 in U1, we compared the expression profile of U1-shCTH with U1-shNT to further understand the link between $H_2S$ biogenesis and HIV-1 latency. Examination of 84 genes related to oxidative stress response revealed differential regulation of 22 genes in U1-shCTH compared with U1-shNT (*Figure 2F* and *Supplementary file 1c*). Interestingly, more than 90% of genes showing altered expression were downregulated in U1-shCTH. Of these, genes encoding key cellular antioxidant enzymes and buffers such as catalase (CAT), superoxide dismutase (SOD1), peroxiredoxin family, thioredoxin family (TXN, TXNRD1 and TXNRD2), sulfiredoxin (SRXN1), glutathione metabolism (GCLM, GSS, and GSR), and sulfur metabolism (MPST, AHCY, and MAT2A) were significantly less expressed in U1-shCTH as compared to U1-shNT (*Figure 2F*). Further, while MPST was downregulated upon CTH knock-down, CBS remained undetectable in U1-shCTH and U1-shNT (*Figure 2F*). Since oxidative stress elicits HIV-1 reactivation (*Bhaskar et al., 2015*), these findings indicate that HIV-1 reactivation through CTH depletion could be a consequence of an altered redox balance. Further, pathways involved in promoting HIV-1 reactivation, such as NF-κB signaling (*Williams et al., 2007*) and apoptosis (*Khan et al., 2015*), were also induced in U1-shCTH as compared to U1-shNT (*Figure 2F*). Notably, multiple HIV-1 restriction factors are important for viral latency including type I interferon signaling (IRF2 and STAT-1) (*Sgarbanti et al., 2002*; *Nguyen et al., 2018*),

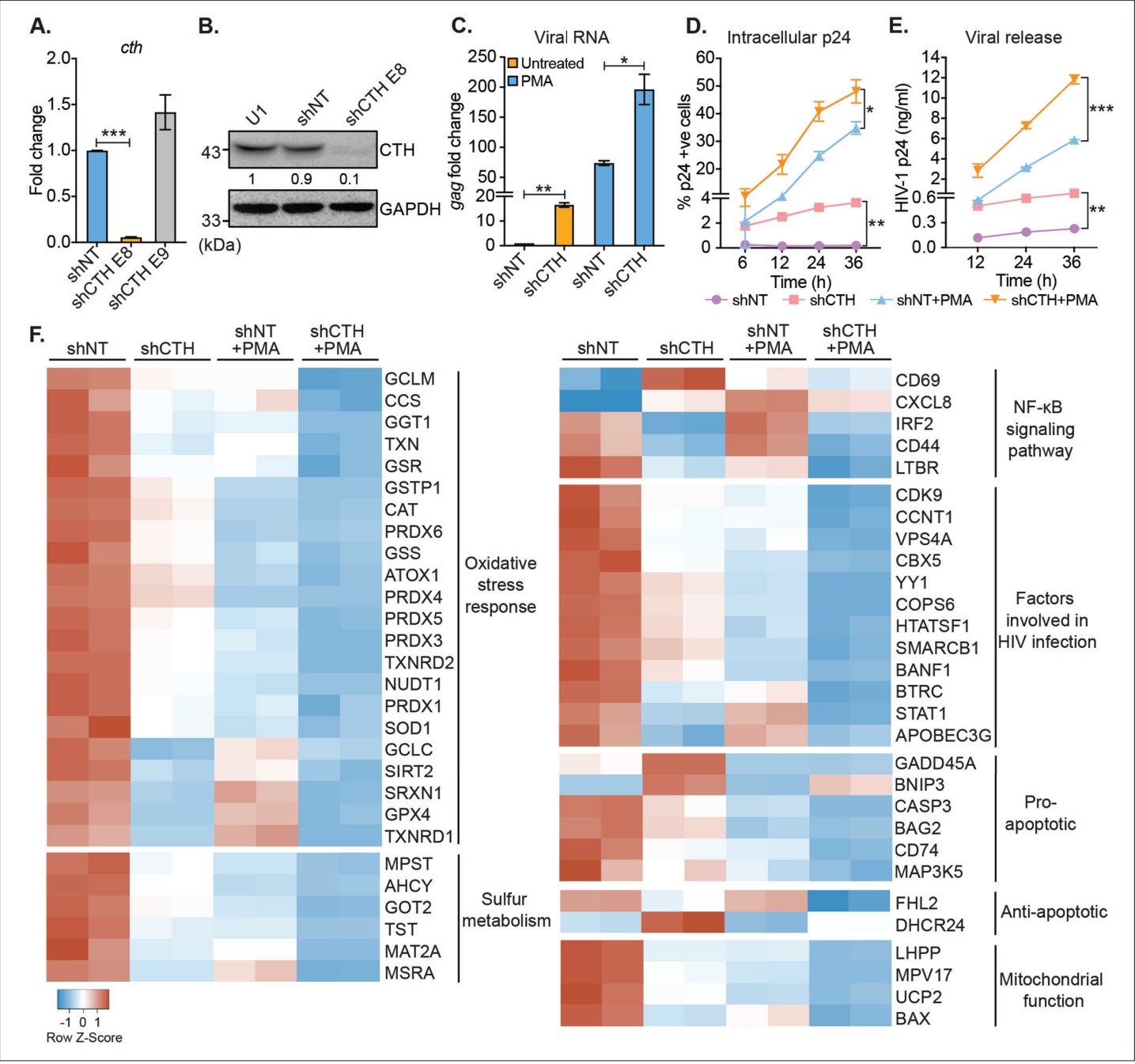

**Figure 2.** Genetic silencing of CTH reactivates HIV-1 and alters gene expression associated with redox stress, apoptosis, and mitochondrial function. (**A**) Total RNA was isolated from shCTH E8, shCTH E9, and non-targeting shRNA (shNT) lentiviral vectors transduced U1 cells and change in CTH mRNA was examined by RT-qPCR. (**B**) Cell lysates of U1, shNT, and shCTH E8 were assessed for CTH abundance using immunoblotting. The CTH band intensities were quantified by densitometric analysis and normalized to GAPDH. (**C**) shCTH and shNT were treated with 5 ng/ml PMA or left untreated for 24 hr and HIV-1 reactivation was determined by *gag* RT-qPCR. (**D, E**) shCTH and shNT were treated with 5 ng/ml PMA or left untreated. At the indicated time points, HIV-1 reactivation was measured by flow-cytometry using fluorescent tagged (PE-labeled) antibody specific to intracellular p24 (Gag) antigen and p24 ELISA in the supernatant. Results are expressed as mean ± standard deviation and are representative of data from two independent experiments. *, p<0.05; **, p<0.01; ***, p<0.001, by two-way ANOVA with Tukey's multiple comparison test. (**F**) Total RNA isolated from untreated or PMA (5 ng/ml; 24 hr)-treated U1-shNT and U1-shCTH were examined by NanoString Technology to assess the expression of genes associated with HIV-1 infection and oxidative stress response. Heatmap showing functional categories of significantly differentially expressed genes. Gene expression data obtained were normalized to internal control β2 microglobulin (B2M), and fold changes were calculated using the nSolver 4.0 software. Genes with fold changes values of >1.5 and p<0.05 were considered as significantly altered. RT-qPCR, reverse transcription quantitative PCR.

The online version of this article includes the following figure supplement(s) for figure 2:

*Figure 2 continued on next page*

*Figure 2 continued*

**Source data 1.** This file contains the source data used to generate *Figure 2B*.

**Source data 2.** This file contains the source data used to make the graphs presented in *Figure 2*.

**Source data 3.** This file contains raw values associated with the NanoString analysis of host genes affected upon depletion of CTH in U1.

**Figure supplement 1.** Genetic silencing of CTH reduces endogenous H$_2$S levels and reactivates HIV-1 from latency.

**Figure supplement 1—source data 1.** This file contains the source data used to generate *Figure 2—figure supplement 1E*.

APOBEC3G (*Gillick et al., 2013*), CDK9-CCNT1 ( *Amini et al., 2002*; *Budhiraja et al., 2013*), SLP-1 (*McNeely et al., 1995*), and chromatin remodelers (SMARCB1, BANF1, and YY1) (*Rafati et al., 2011*; *Coull et al., 2000*) were downregulated in U1-shCTH. HIV-1 proteins are known to target mitochondria to induce mitochondrial depolarization, elevate mitochondrial reactive oxygen species (ROS), and apoptosis for replication (*Badley et al., 2003*). Several genes involved in sustaining mitochondrial function and membrane potential (e.g., BAX, UCP2, LHPP, and MPV17) were repressed in U1-shCTH (*Figure 2F*), highlighting a potential association between unrestricted virus replication, mitochondrial dysfunction, and CTH depletion.

Since PMA is known to reactivate HIV-1 (*Pardons et al., 2019b*), we examined the gene expression in U1-shNT upon reactivation of HIV-1 by PMA. We found that 80% of genes affected by the depletion of CTH were similarly perturbed in response to PMA. Signature of transcripts associated with oxidative stress response, sulfur metabolism, anti-viral factors, and mitochondrial function was comparable in U1-shCTH and PMA-treated U1-shNT (*Figure 2F*). Based on these similarities, we hypothesized that combining PMA with CTH depletion would have an additive effect on the expression of genes linked to HIV-1 reactivation. Indeed, exposure of U1-shCTH to PMA induced gene expression changes, which surpassed those produced by PMA treatment or CTH-depletion alone (*Figure 2F*). Notably, significant expression changes in U1-shCTH upon PMA treatment are consistent with our data showing maximal HIV-1 activation under these conditions (see *Figure 2C*). Overall, these results indicate that CTH contributes to HIV-1 latency by modulating multiple pathways coordinating cellular homeostasis (e.g., redox balance and mitochondrial function) and the anti-viral response.

## CTH is required to maintain redox homeostasis and mitochondrial function

Physiological levels of H$_2$S support redox balance and mitochondrial function by maintaining GSH balance (*Kimura et al., 2010*), protecting against ROS (*Kimura et al., 2010*), and acting as a substrate for the electron transport chain (ETC) (*Modis et al., 2014*; *Fu et al., 2012*). On this basis, we reasoned that the depletion of CTH could contribute to redox imbalance and mitochondrial dysfunction, both of which are known to promote HIV-1 reactivation (*Tyagi et al., 2020*; *Bhaskar et al., 2015*; *Singh et al., 2021*). We first measured total glutathione content (GSH+GSSG) and GSSG concentration in U1-shNT and U1-shCTH using chemical-enzymatic analysis of whole-cell extract (*Bhaskar et al., 2015*). Whole-cell glutathione content was not significantly different in U1-shNT and U1-shCTH (p=0.2801) (*Figure 3A*). However, the GSSG concentration was elevated in U1-shCTH resulting in a concomitant decrease in the GSH/GSSG ratio of U1-shCTH as compared to U1-shNT (p=0.0032) (*Figure 3A*). The increased GSSG pool and reduced GSH/GSSG poise confirm that cells are experiencing oxidative stress upon CTH depletion. We measured total ROS using a fluorescent probe, 5,6-chloromethyl-2′, 7′-dichlorodihydrofluorescein diacetate (CM-H$_2$DCFDA), which non-specifically responds to any type of ROS within the cells. Both U1-shNT and U1-shCTH showed comparable levels of cytoplasmic ROS (*Figure 3—figure supplement 1A*), which remained unchanged after stimulation with PMA. In addition to cytoplasmic ROS, we also measured mitochondrial ROS (mitoROS) using the red fluorescent dye MitoSOX, which selectively stains mitoROS. Lowering the levels of CTH severely increased mitoROS in U1, which was further accentuated after PMA stimulation (*Figure 3B*). We then studied the effect of CTH depletion on mitochondrial functions using a Seahorse XF Extracellular Flux Analyzer (Agilent) as described previously by us (*Figure 3C*; *Tyagi et al., 2020*; *Szabo et al., 2014*). Both basal and ATP-coupled respiration was significantly decreased in U1-shCTH as compared to U1-shNT (*Figure 3D–E*), consistent with the role of endogenous H$_2$S in reducing cytochrome C oxidase for respiration (*Szabo et al., 2014*). The maximal respiratory capacity, attained by the dissipation of the mitochondrial proton gradient with the uncoupler FCCP, was markedly diminished in U1-shCTH (*Figure 3E*). The maximal

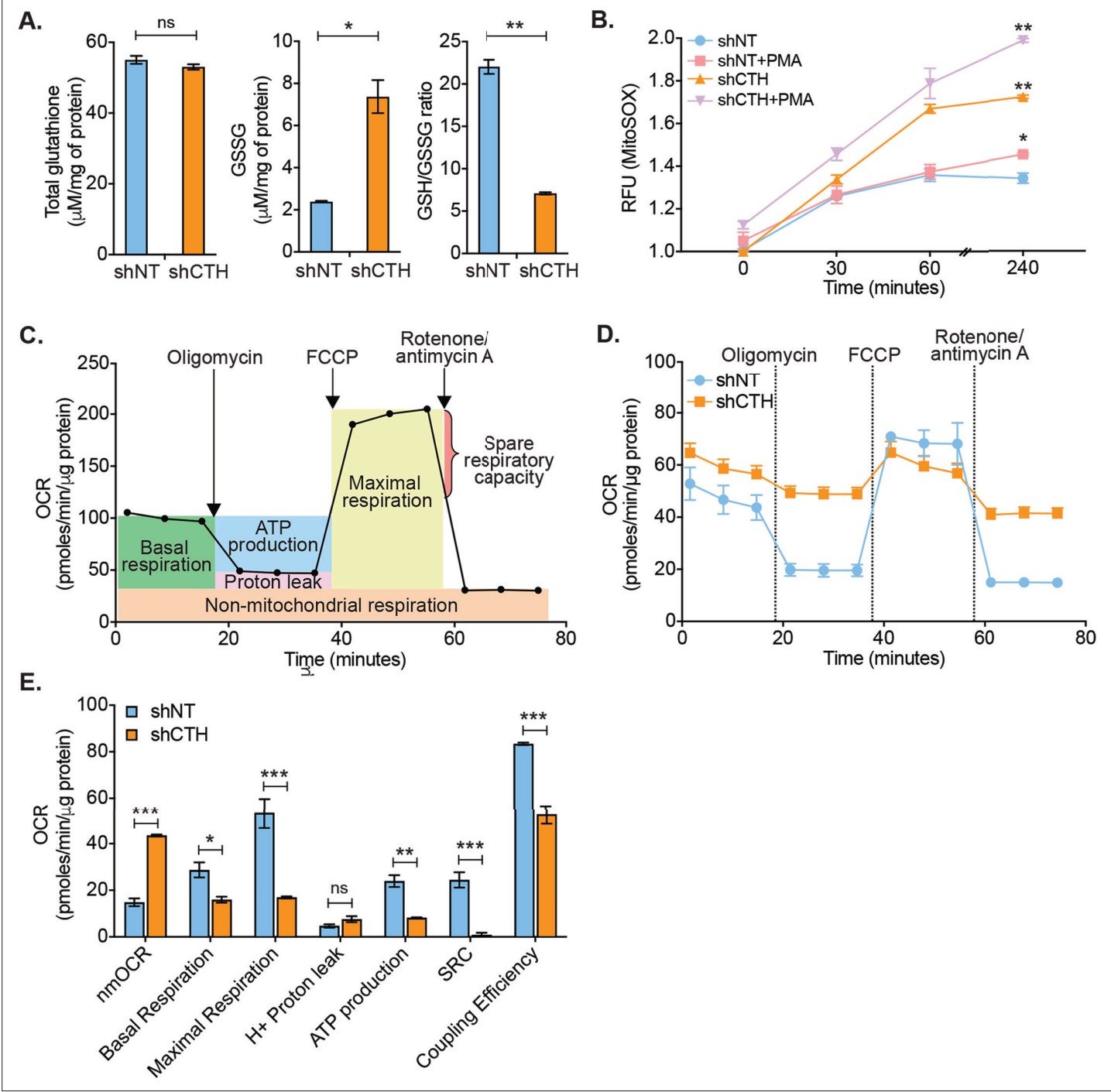

**Figure 3.** CTH maintains redox homeostasis and mitochondrial bioenergetics to promote HIV-1 latency. (**A**) Total and oxidized cellular glutathione (GSSG) content was assessed in U1-shCTH and U1-shNT cell lysates using glutathione assay kit. (**B**) U1-shNT and U1-shCTH were stained with MitoSOX Red dye for 30 min at 37°C and analyzed by flow cytometry (Ex-510 nm, Em-580 nm). (**C**) Schematic representation of Agilent Seahorse XF Cell Mito Stress test profile to assess key parameters related to mitochondrial respiration. (**D**) U1-shNT and U1-shCTH ($5\times10^4$) were seeded in triplicate wells of XF microplate and incubated for 1 hr at 37°C in a non-$CO_2$ incubator. Oxygen consumption was measured without adding any drug (basal respiration), followed by measurement of OCR change upon sequential addition of 1 μM oligomycin (ATP synthase inhibitor) and 0.25 μM carbonyl cyanide 4-(trifluoromethoxy) phenylhydrazone (FCCP), which uncouples mitochondrial respiration and maximizes OCR. Finally, rotenone (0.5 μM) and antimycin A (0.5 μM) were injected to completely inhibit respiration by blocking complex I and complex III, respectively. (**E**) Various respiratory parameters derived from OCR measurement were determined by Wave desktop software. nmOCR; non-mitochondrial oxygen consumption rate and SRC; spare respiratory capacity. Error bar represents standard deviations from mean. Results are representative of data from three independent experiments. *, $p<0.05$; **,

*Figure 3 continued on next page*

*Figure 3 continued*

p<0.01; ***, p<0.001; ns, non-significant, by two-way ANOVA with Bonferroni's multiple comparison test.

The online version of this article includes the following figure supplement(s) for figure 3:

**Source data 1.** This file contains the source data used to make the graphs presented in *Figure 3*.

**Figure supplement 1.** Effect of CTH knockdown on cytosolic ROS generation.

respiration also facilitated the estimation of the spare respiratory capacity (SRC), which was nearly exhausted in U1-shCTH. Additional hallmarks of HIV-1 reactivation and dysfunctional mitochondria such as coupling efficiency and non-mitochondrial oxygen consumption rate (nmOCR) (*Tyagi et al., 2020*) were also adversely affected upon depletion of CTH (*Figure 3E*). These results indicate that CTH depletion decelerates respiration, diminishes the capacity of macrophages to maximally respire, and promotes nmOCR. All of these parameters are important features of mitochondrial health and are likely to be crucial for maintaining HIV-1 latency.

## A small-molecule H₂S donor diminished HIV-1 reactivation and viral replication

Having shown that diminished levels of endogenous $H_2S$ are associated with redox imbalance, mitochondrial dysfunction, and reactivation of HIV-1 from latency, we next examined if elevating $H_2S$ levels using a small-molecule $H_2S$ donor- GYY4137 sustains HIV-1 latency. The GYY4137 is a widely used $H_2S$ donor that releases a low amount of $H_2S$ over a prolonged period to mimic physiological production (*Li et al., 2008*). Because $H_2S$ discharge is unusually sluggish by GYY4137, the final concentration of $H_2S$ released is likely to be significantly lower than the initial concentration of GYY4137. We confirmed this by measuring $H_2S$ release in U1 cells treated with low (0.3 mM) or high (5 mM) concentrations of GYY4137 using methylene blue colorimetric assay. As a control, we treated U1 cells with 0.3 mM NaHS, which rapidly releases a high amount of $H_2S$. As expected, NaHS treatment rapidly released 90% of $H_2S$ within 5 min of treatment (*Figure 4A*). In contrast, GYY4137 uniformly released a low amount of $H_2S$ for the entire 120 hr duration of the experiment (*Figure 4A*).

We systematically tested the effect of GYY4137 on HIV-1 reactivation using multiple models of HIV-1 latency and replication. As a control, we used N-acetyl cysteine (NAC) that is known to block HIV-1 reactivation (*Roederer et al., 1990*). Pretreatment of U1 with a non-toxic dose of GYY4137 (5 mM) (*Figure 4—figure supplement 1A*) diminished the expression of *gag* transcript by twofold at 24 hr and tenfold at 48 hr post-PMA treatment (*Figure 4B*). The effect of GYY4137 on p24 levels in the supernatant was even more striking as it completely abolished the time-dependent increase in p24 concentration post-PMA treatment (*Figure 4C*). As expected, pretreatment with NAC similarly prevented PMA-triggered reactivation of HIV-1 in U1 (*Figure 4C*). Pretreatment of U1 with spent GYY4137, which comprises the decomposed backbone, showed no effect on PMA induced *gag* transcript (*Figure 4—figure supplement 1B*). Because TNF-α is a physiologically relevant cytokine that reactivates HIV-1 from latency (*Folks et al., 1987*), we tested the effect of GYY4137 on TNF-α-mediated virus reactivation. Treatment of U1 with TNF-α stimulated the expression of *gag* transcript by 42-fold and 169-fold at 24 and 48 hr, post-treatment, respectively (*Figure 4D*). The addition of GYY4137 or NAC nearly abolished the reactivation of HIV-1 in response to TNF-α treatment (*Figure 4D*). Earlier, we showed that depletion of CTH stimulated HIV-1 reactivation in U1 (see *Figure 2C*). Therefore, we tested if GYY4137 could complement this genetic deficiency and subvert HIV-1 reactivation. The U1-shCTH cells were pre-treated with 5 mM GYY4137 and p24 levels in the supernatant were measured. The elevated levels of p24 in U1-shCTH were reduced by twofold under basal conditions and 3.2-fold upon PMA stimulation in response to GYY4137 (*Figure 4E*). Both RNAi and chemical complementation data provide evidence that $H_2S$ is one of the factors regulating HIV-1 latency and reactivation in U1.

Similar to U1, we next examined whether GYY4137 subverts HIV-1 reactivation in the J1.1 T-cell line model. Treatment of J1.1 with PMA for 12 hr resulted in a 38-fold increase in *gag* transcript as compared to untreated J1.1 (*Figure 4F*), indicating efficient reactivation of HIV-1. Importantly, pretreatment with various non-toxic concentrations of GYY4137 (*Figure 4—figure supplement 1C*) reduced the stimulation of *gag* transcription by twofold upon subsequent exposure to PMA (*Figure 4F*). Interestingly, low concentration of GYY4137 (0.3 mM) was relatively more effective than high concentration (5 mM)

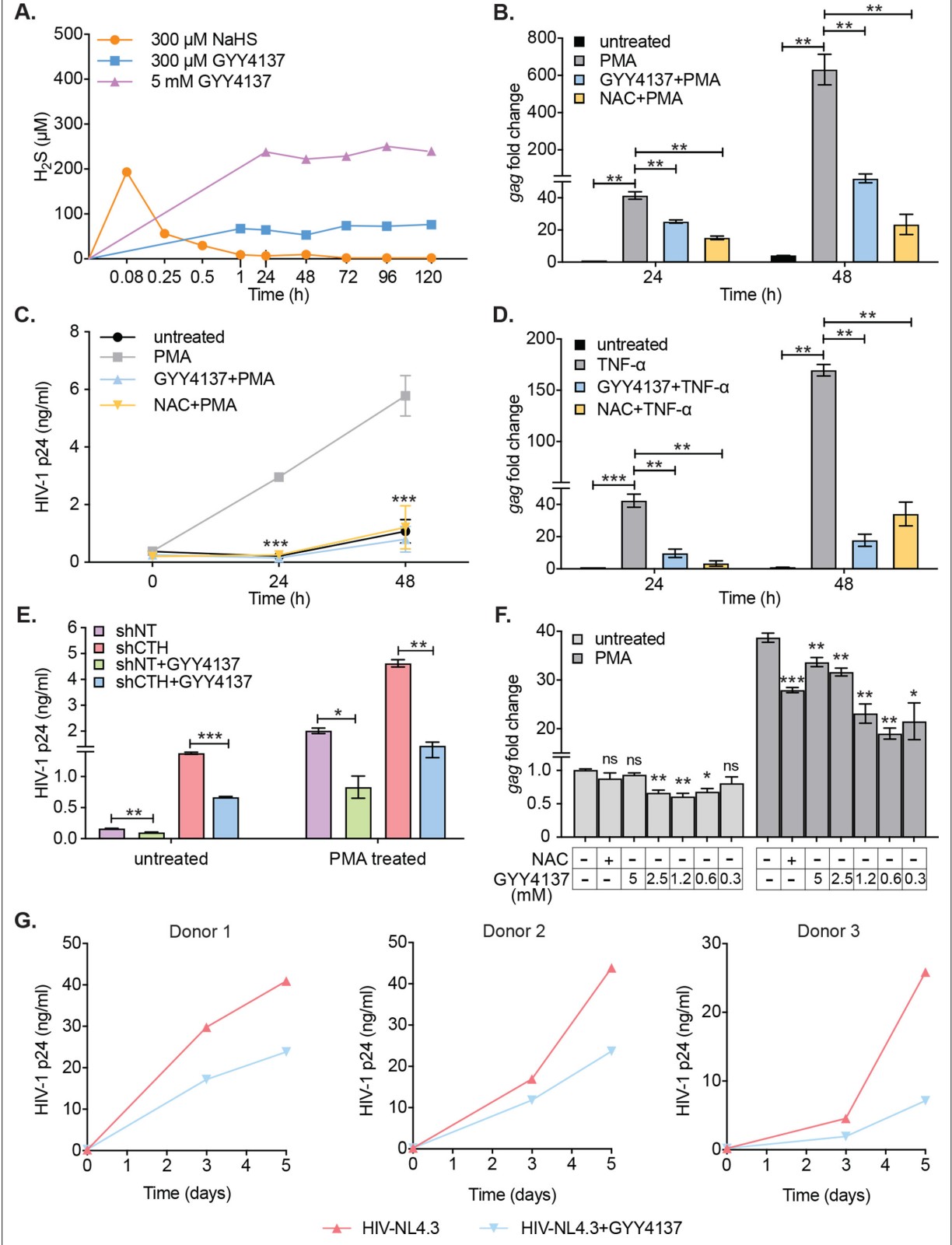

**Figure 4.** H$_2$S donor (GYY4137) suppresses HIV-1 reactivation and replication. (**A**) U1 cells were treated with NaHS or GYY4137 and media supernatant was harvested to assess H$_2$S production by methylene blue assay over time. (**B, C**) U1 cells were pre-treated with 5 mM GYY4137 or 5 mM NAC for 24 hr and then stimulated with 5 ng/ml PMA for 24 and 48 hr. Total RNA was isolated and HIV-1 reactivation was assessed by *gag* RT-qPCR (**B**). Culture supernatant was harvested to monitor HIV-1 release by p24 ELISA (**C**). (**D**) U1 cells were pretreated with 5 mM GYY417, 5 mM NAC for 24 hr or left

*Figure 4 continued on next page*

*Figure 4 continued*

untreated and then stimulated with 100 ng/ml TNF-α for 24 and 48 hr. HIV-1 reactivation was assessed by *gag* RT-qPCR. (**E**) U1-shCTH and U1-shNT were pretreated with 5 mM GYY4137 for 24 hr and stimulated with 5 ng/ml PMA for 24 hr. Culture supernatant was harvested to determine HIV-1 reactivation by HIV-1 p24 ELISA. (**F**) J1.1 cells were pretreated with indicated concentrations of GYY4137 or 5 mM NAC for 24 hr and then stimulated with 5 ng/ml PMA for 12 hr. Cells were harvested to isolate total RNA and HIV-1 reactivation was assessed by *gag* RT-qPCR. (**G**) Primary human CD4$^+$ T cells purified from PBMCs samples of healthy donors were activated with anti-CD3/anti-CD28 beads for 3 days. Activated primary CD4$^+$ T cells (three healthy donors) were pre-treated with 800 μM GYY4137 for 6 hr, and infected with HIV-NL4.3 (1 ng p24/10$^6$ cells). Post-infection (p.i.) cells were washed, seeded in fresh media, and treated with 800 μM GYY4137 or left untreated. Virus released in the supernatant was quantified by p24 ELISA at 3$^{rd}$ and 5$^{th}$ day post-infection. Results are expressed as ± standard deviation and data are representative of three independent experiments. *, p<0.05; **, p<0.01; ***, p<0.001, ns, non-significant by two-way ANOVA with Tukey's multiple comparison test. PBMC, peripheral blood mononuclear cell; RT-qPCR, reverse transcription quantitative PCR.

The online version of this article includes the following figure supplement(s) for figure 4:

**Source data 1.** This file contains the source data used to make the graphs presented in *Figure 4*.

**Figure supplement 1.** Effect of GYY4137 on HIV-1 reactivation and cellular viability.

in subverting HIV-1 reactivation. While intriguing, these findings are consistent with the reports that high concentrations of H$_2$S induce pro-oxidant effects on cellular physiology (*Olas et al., 2019*), which might have dampened the suppressive action of GYY4137 on HIV-1 transcription in J1.1. However, this needs further experimentation. The inhibitory effect of GYY4137 on HIV-1 reactivation was relatively greater compared to NAC in J1.1 (*Figure 4F*). Consistent with U1 and J1.1, treatment of J-Lat with 5 μM prostratin for 24 hr induced significant HIV-1 reactivation, which was translated as 100% increase in GFP$^+$ cells (*Figure 4—figure supplement 1D*). Pretreatment with GYY4137 significantly reduced HIV-1 reactivation in a dose-dependent manner (*Figure 4—figure supplement 1D*).

We also examined if an endogenous increase in H$_2$S by GYY4137 halts the replication of HIV-1 in primary CD4$^+$ T cells. To this end, we infected primary CD4$^+$ T cells isolated from peripheral blood mononuclear cells (PBMCs) of three healthy human donors with HIV-1 (pNL4.3) and monitored HIV-1 replication by measuring the levels of p24 in the supernatant at 3 and 5 days post-infection. The p24 ELISA confirmed a time-dependent increase in viral load, which was uniformly reduced upon pretreatment with a non-toxic concentration of GYY4137 (0.8 mM) (*Figure 4G* and *Figure 4—figure supplement 1E*). Overall, these data establish that elevated levels of endogenous H$_2$S efficiently suppress HIV-1 reactivation and replication.

## GYY4137 reduced the expression of host genes involved in HIV-1 reactivation

To dissect the mechanism of GYY4137-mediated inhibition on HIV-1 reactivation, we examined the expression of 185 genes associated with HIV-1 reactivation using the NanoString nCounter Technology as described above (*Figure 5A* and *Supplementary file 1d*). Expression was analyzed for viral latency (unstimulated U1), reactivation (PMA-stimulated treated U1; PMA), and H$_2$S-mediated suppression (U1-GYY+PMA). Consistent with the role of PMA-mediated oxidative stress in preceding HIV-1 reactivation (*Bhaskar et al., 2015*), expression of genes encoding ROS, and RNS generating enzymes (e.g., NADPH oxidase [NCF1 and CYBB] and Nitric oxide synthase [NOS2]) were upregulated (*Figure 5A*). Also, the expression of major antioxidant enzymes (e.g., GPXs, PRDXs, and CAT) and redox buffers (GSH and TRX pathways) remained repressed in U1-PMA, indicative of elevated oxidative stress. Additionally, pro-inflammatory signatures (e.g., TGFB1 and SERPINA1) and trans-activators (e.g., FOS) were induced, whereas host factors involved in HIV-1 restriction (e.g., IRF1 and YY1) were repressed in U1-PMA as compared to U1. In agreement with the reduction in endogenous H$_2$S levels during HIV-1 reactivation, expression of CTH involved in H$_2$S anabolism was downregulated and H$_2$S catabolism (SQRDL) was upregulated by PMA.

We noticed that the treatment with GYY4137 reversed the effect of PMA on the expression of genes associated with oxidative stress, inflammation, anti-viral response, apoptosis, and trans-activators (*Figure 5A*). For example, GYY4137 elicited a robust induction of genes regulated by nuclear factor erythroid 2-related factor 2 (Nrf2) in U1-GYY+PMA compared to U1 or U1-PMA (*Figure 5A*). Nrf2 acts as a master regulator of redox metabolism (*Pall and Levine, 2015*) by binding to the antioxidant response element (ARE) and initiating transcription of major antioxidant genes (*e.g.*, GSH pathway, TXNRD1, HMOX1, GPX4, and SRXN1). Interestingly, while the expression of Nrf2-dependent CTH

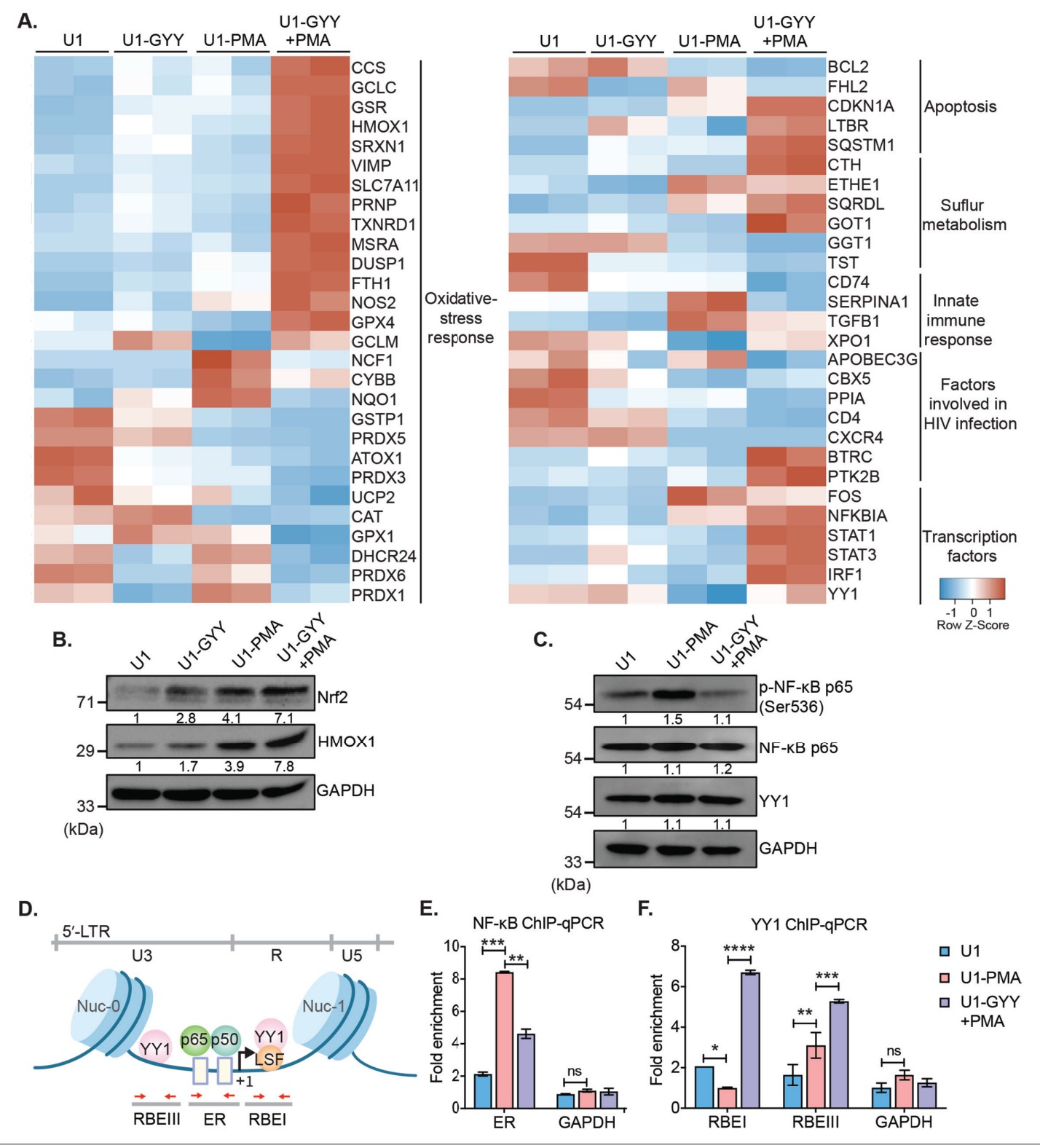

**Figure 5.** GYY4137 modulates Nrf2, NF- $\kappa$ B, and YY1 pathways. (**A**) U1 cells pre-treated with 5 mM GYY4137 for 24 hr or left untreated and then stimulated with 5 ng/ml PMA for 24 hr or left unstimulated. Total RNA was isolated and expression of genes associated with HIV-1 infection and oxidative stress response was assessed by nCounter NanoString Technology. Heatmap showing functional categories of significant DEGs in all four conditions: untreated (U1), GYY4137 alone (U1-GYY) or PMA- alone (U1-PMA), and GYY4137+PMA (U1-GYY+PMA). Gene expression data obtained were normalized to internal control $\beta_2$ microglobulin (B2M), and fold changes were calculated using the nSolver 4.0 software. Genes with fold changes

*Figure 5 continued on next page*

*Figure 5 continued*

values of >1.5 and p<0.05 were considered as significantly altered. (**B**) Total cell lysates were used to analyze the expression levels of Nrf2 and HMOX1 by immunoblotting. Results were quantified by densitometric analysis of Nrf2 and HMOX1 band intensities and normalized to GAPDH. (**C**) U1 cells were pretreated with 5 mM GYY4137 for 6 hr and then stimulated with 30 ng/ml PMA for 4 hr or left unstimulated. Cells were harvested to prepare total cell lysate. Levels of phosphorylated NF-$\kappa$B p65 (Ser536), NF-$\kappa$B p65, and YY1 were determined by immunoblotting. Results were quantified by densitometric analysis for each blot and were normalized to GAPDH. (**D**) Schematic depiction of the binding sites for NF-$\kappa$B (p65-p50 heterodimer) and YY1 on HIV-1 5′-LTR. Highlighted arrow in red indicates the regions targeted for genomic qPCR; ER site for NF-$\kappa$B, and RBEI and RBEIII sites for YY1 enrichments, respectively. (**E, F**) U1 cells were pretreated with 5 mM GYY4137 for 6 hr, stimulated with PMA (30 ng/ml) for 4 hr, fixed with formaldehyde, and lysed. Lysates were subjected to immunoprecipitation for p65 and YY1 and protein-DNA complexes were purified using protein-G magnetic beads. The enrichment of NF-$\kappa$B p65 and YY1 on HIV-1 LTR was assessed by qPCR for designated regions using purified DNA as a template. Results are expressed as mean ± standard deviation and data are representative of three independent experiments *, p<0.05; **, p<0.01; ***, p<0.001; ****, p<0.0001; ns, non-significant, by two-way ANOVA with Tukey's multiple comparison test. DEG, differentially expressed gene.

The online version of this article includes the following figure supplement(s) for figure 5:

**Source data 1.** This file contains raw values associated with the NanoString analysis of host genes affected upon PMA induced HIV reactivation in presence or absence of GYY4137 treatment.

**Source data 2.** This file contains the source data used to generate *Figure 5B*.

**Source data 3.** This file contains the source data used to generate *Figure 5C*.

**Source data 4.** This file contains the source data used to make the graphs presented in *Figure 5*.

gene was induced in U1-GYY+PMA, MPST remained unaffected. The Nrf2 activity has been shown to pause HIV-1 infection by inhibiting the insertion of reverse-transcribed viral cDNA into the host chromosome (*Furuya et al., 2016*). Furthermore, sustained activation of Nrf2-dependent antioxidant response is essential for the establishment of viral latency (*Shytaj et al., 2020*). A few genes encoding $H_2O_2$ detoxifying enzymes (e.g., CAT, GPX1, and PRDXs) were repressed in U1-GYY+PMA, indicating that the expression of these enzymes was likely counterbalanced by the elevated expression of other antioxidant systems by $H_2S$. In line with the antagonistic effect of GYY4137 on HIV-1 transcription, the expression of HIV-1 trans-activator (FOS) was downregulated, and an inhibitor of NF-κB signaling (NFKBIA) was induced in U1-GYY+PMA as compared to U1 or U1-PMA. Finally, GYY4137 stimulated the expression of several anti-HIV (YY1, STAT1, STAT3, and IRF1) and pro-survival factors (CDKN1A and LTBR) that were repressed by PMA (*Figure 5A*). Altogether, $H_2S$ supplementation induces a major realignment of redox metabolism and immune pathways associated with HIV-1 reactivation.

## GYY4137-mediated modulation of the Keap1-Nrf2 axis, NF-κB signaling, and activity of epigenetic factor YY1

Our expression data indicate activation of the Nrf2 pathway and modulation of transcription factors such as NF-κB and YY1 upon treatment with GYY4137. We tested if the mechanism of $H_2S$-mediated subversion of HIV-1 reactivation involves these pathways. $H_2S$ has recently been shown to prevent cellular senescence by activation of Nrf2 via S-persulfidation of its negative regulator Keap-1 (*Yang et al., 2013*). Under unstimulated conditions, Nrf2 binds to Keap1, and the latter promotes Nrf2 degradation via the proteasomal machinery (*Yang et al., 2013*). Nrf2 disassociates from the S-persulfidated form of Keap1, accumulates in the cytoplasm, and translocates to nuclei where it induces transcription of antioxidant genes upon oxidative stress (*Yang et al., 2013*). We first examined if GYY4137 treatment accumulates Nrf2 in the cytoplasm. As expected, Nrf2 was not detected in U1 owing to its association with Keap1 under unstimulated conditions. However, a noticeable accumulation of Nrf2 was observed in U1-GYY+PMA compared to U1 or U1-PMA (*Figure 5B*). As an additional verification, we quantified the levels of an Nrf2-dependent protein HMOX-1. Similar to Nrf2, the levels of HMOX-1 were also induced in U1-GYY+PMA (*Figure 5B*). These findings are consistent with our NanoString data showing activation of Nrf2-specific oxidative stress responsive genes in GYY4137 treated U1.

Next, we determined if GYY4137 treatment targets NF-κB, which is a major regulator of HIV-1 reactivation (*Williams et al., 2007*). The cellular level of phosphorylated serine-536 in p65, a major subunit of NF-κB, is commonly measured to assess NF-κB activation (*Zhong et al., 1998*). Since PMA reactivates HIV-1, we observed an increase in p65 ser-536 phosphorylation in U1-PMA (*Figure 5C*). In contrast, pretreatment with GYY4137 significantly decreased PMA-induced p65 ser-536 phosphorylation (*Figure 5C*). These findings suggest that GYY4137 is likely to affect the DNA binding and transcriptional activity of NF-κB. Consistent with this idea, a two-step chromatin immunoprecipitation and

genomic qPCR (ChIP-qPCR) assay showed that GYY4137-treatment significantly reduced the occupancy of p65 at its binding site (ER, enhancer region) on the HIV-1 LTR (*Figure 5D–E*).

GYY4137 induces the expression of another transcription factor YY1, which binds to HIV-1 LTR at RBEI and RBEIII and recruits histone deacetylase (HDACs) to facilitate repressive chromatin modifications (*Coull et al., 2000*). Overexpression of YY1 is known to promote HIV-1 latency (*Bernhard et al., 2013*). We tested if GYY4137 promotes the binding of YY1 at RBEI and RBEIII sites on HIV-1 LTR by ChIP-qPCR. The occupancy of YY1 at RBEI and RBEIII was significantly enriched in the case of U1-GYY+PMA as compared to U1-PMA or U1 (*Figure 5F*). Taken together, these results indicate that increasing endogenous $H_2S$ levels by using a slow-releasing donor effectively modulate the Keap1-Nrf2 axis and activity of transcription factors to maintain redox balance and control HIV-1 reactivation.

## GYY4137 blocks HIV-1 rebound from latent CD4⁺ T cells isolated from infected individuals on suppressive ART

Having shown that $H_2S$ suppresses HIV-1 reactivation in multiple cell line models of latency, we next studied the ability of GYY4137 to limit virus transcription in primary CD4⁺ T cells derived from virally suppressed patients. We used a previously established methodology of maintaining CD4⁺ T cells from infected individuals on ART for a few weeks without any loss of phenotypic characteristics associated with HIV-1 reservoirs (*Kessing et al., 2017*). We isolated CD4⁺ T cells from PBMCs of five HIV-1 infected subjects on suppressive ART. The CD4⁺ T cells were expanded in the presence of interleukin-2 (IL-2), phytohemagglutinin (PHA), and feeder cells (*Figure 6A*). The expanded cells were cultured for 4 weeks in a medium containing IL-2 with ART (100 nM efavirenz, 180 nM zidovudine, and 200 nM raltegravir) either in the presence or absence of GYY4137 (100 µM) (*Figure 6A*). In this model, the virus reactivates by day 14 followed by progressive suppression at day 21 and latency establishment by day 35. The virus can be reactivated from this latent phase using well-known LRAs (*Kessing et al., 2017*). We quantified the temporal expression of viral RNA using reverse transcription quantitative PCR (RT-qPCR) periodically at 7-day interval. The levels of viral RNA increased initially (days 7–14), followed by low to undetectable levels by day 28 (*Figure 6B*). By day 28, virus RNA was uniformly untraceable in cells derived from GYY4137 treated and untreated groups, indicating the establishment of latency (*Figure 6B*).

We next assessed the ability of GYY4137 to efficiently block viral reactivation. On day 28, we stimulated the CD4⁺ T cells with the protein kinase C (PKC) activator, prostratin, in the absence of any treatment. The activation of viral transcription was measured 24 hr later by RT-qPCR. The removal of ART uniformly resulted in a viral reactivation by prostratin in CD4⁺ T cells of HIV1 patients (*Figure 6C*). In contrast, upon ART+GYY4137 removal followed by prostratin stimulation, viral reactivation was attenuated by 90% (N=5, mean) for all five patient samples, and individual inhibition ranged from 78.9% to 100% (*Figure 6C*). Overall, pretreatment with ART+GYY4137 significantly reduced prostratin-mediated HIV-1 reactivation when compared to ART alone (p<0.01) (*Figure 6D*).

We also examined the immune-phenotype of CD4⁺ T cells treated with ART or ART+GYY4137 by monitoring the expression of activation (CD38) and quiescence (CD127) marker. As expected, CD38 expression increased, and CD127 expression decreased at days 7 and 14 during the activation phase, followed by a gradual reversal of the pattern during the resting phase at days 21 and 28 (*Figure 6E–F* and *Figure 6—figure supplement 1A*). Interestingly, GYY4137 treatment did not alter the temporal changes in the expression of activation and quiescence markers on CD4⁺ T cells (*Figure 6E–F*). Since memory CD4⁺ T cells are preferentially targeted by HIV (*Chomont et al., 2009*; *Brenchley et al., 2004*), we further analyzed if the status of naive ($T_N$), central memory ($T_{CM}$), transitional memory ($T_{TM}$), and effector memory ($T_{EM}$) is affected by the GYY4137 in our ex vivo expansion model. We observed that the CD4⁺ T cells that responded to ex vivo activation were mainly composed of $T_{TM}$ and $T_{EM}$ in our patient samples (*Figure 6—figure supplement 1B-E*). This is consistent with the study reporting the presence of translation-competent genomes mainly in the $T_{TM}$ and $T_{EM}$ of ART-suppressed individuals (*Pardons et al., 2019a*). Interestingly, the fraction of $T_{TM}$ shows a progressive decline, whereas $T_{EM}$ displays gradual increase during transition from activation (7–14 days) to quiescence phase (21–28 days) (*Figure 6—figure supplement 1D-E*). The addition of GYY4137 did not affect dynamic changes in the frequency of $T_{TM}$ and $T_{EM}$ subpopulations (*Figure 6—figure supplement 1D-E*).

Finally, we tested if the reduction in viral RNA upon GYY4137 treatment is due to the loss of cells with the ability to reactivate virus or selection of cells subset that is non-responsive to reactivating

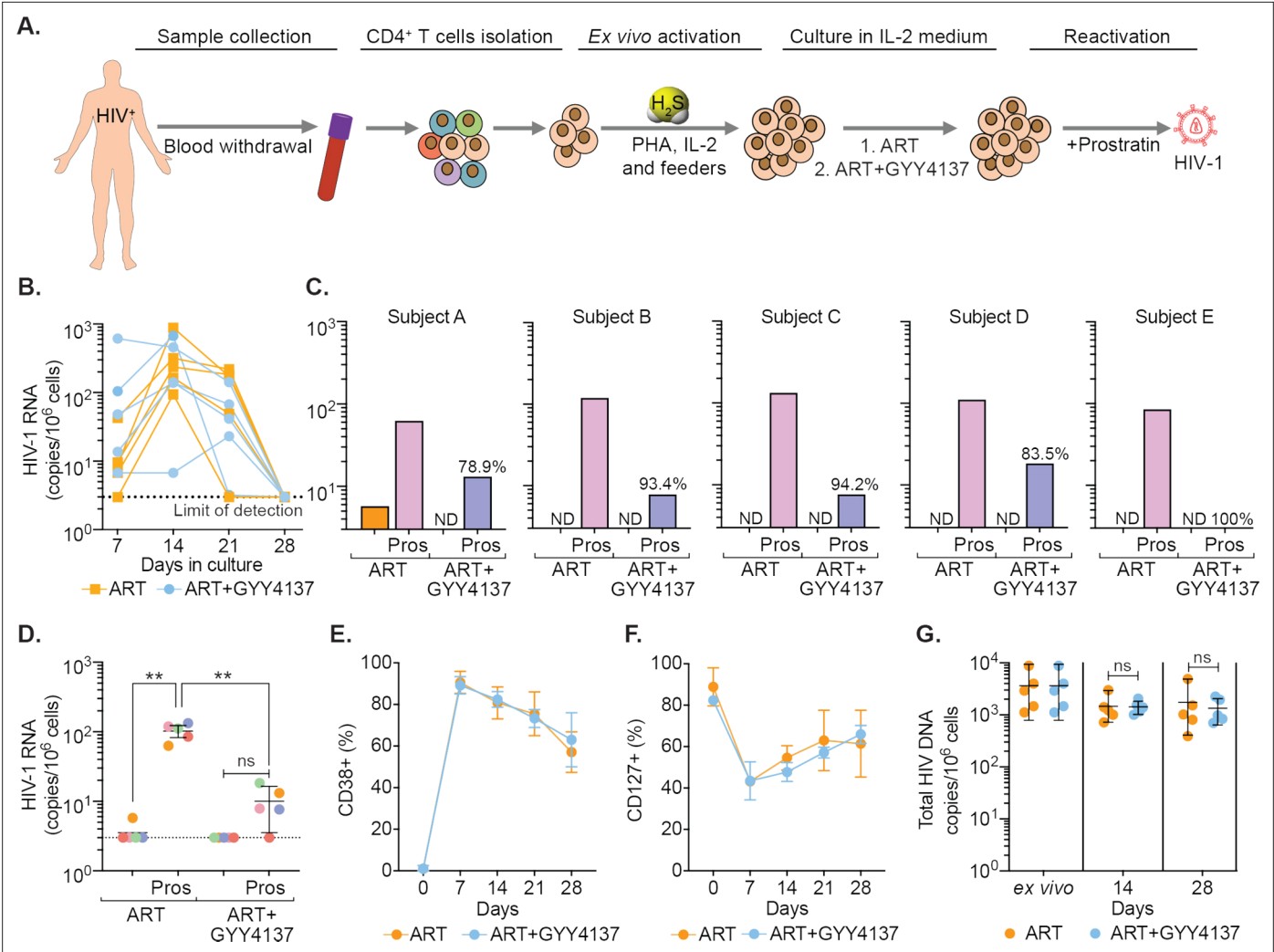

**Figure 6.** GYY4137 subverts HIV-1 reactivation in latent CD4+ T cells derived from HIV-1 infected patients. (**A**) Schematic representation of the experimental design to study the effect of H$_2$S on HIV-1 reactivation using CD4+ T cells from ART-treated HIV-1 infected subjects. CD4+ T cells were sorted and activated with PHA (1 μg/ml), IL-2 (100 U/ml), feeder PBMCs (gamma-irradiated) from healthy donors. CD4+ T cells were activated in the presence of ART or ART in combination with 100 μM GYY4137. Post-activation cells were cultured with ART alone or ART +GYY4137 treatment in IL-2 containing medium. (**B**) Total RNA was isolated from five patients CD4+ T cells expanded in ART or ART+GYY4137. HIV-1 RNA levels were measured every 7 days by ultrasensitive semi-nested RT-qPCR with detection limit of three viral RNA copies per million cells. (**C**) On day 28, cells were washed of any treatment and both ART or ART+GYY4137 treatment groups were stimulated with 1 μM prostratin for 24 hr. HIV-1 RNA copies were assessed by RT-qPCR. Reduction in viral stimulation in GYY4137 treated samples is represented as percentage values. ND, non-determined. (**D**) Aggregate plot for five patients from data shown in (**C**). (**E, F**) Primary CD4+ T cells expanded and cultured in the presence of ART or ART+GYY4137 were analyzed over time for the expression of activation (CD38) and quiescence markers (CD127) by flow cytometry. (**G**) Total HIV-1 DNA content was determined up to 28 days in ART or ART+GYY4137 treated groups. Results are expressed as mean ± standard deviation. **, p<0.01; ns, non-significant, by two-way ANOVA with Tukey's multiple comparison test. RT-qPCR, reverse transcription quantitative PCR.

The online version of this article includes the following figure supplement(s) for figure 6:

**Source data 1.** This file contains the source data used to make the graphs presented in *Figure 6*.

**Figure supplement 1.** Phenotypic features of CD4+ T cells were preserved upon prolonged treatment with GYY4137.

**Figure supplement 2.** Lack of HIV-1 reactivation in CD4+ T cells cultured in the presence of GYY4137 is not due to impaired T cell activation.

**Figure supplement 3.** Flow cytometry gating strategy for ex vivo stimulation of PBMCs derived from ART-treated HIV-1 patients cultured in the presence of GYY4137 in combination with ART or ART alone.

stimuli. We did not find significant differences in total HIV-1 DNA content between the CD4$^+$ T cells immediately isolated from the patient's PBMCs (ex vivo) and the expanded cells treated with ART+ GYY4137 or ART alone for the entire duration of the experiment (*Figure 6G*). The viability of cells remained comparable between ART+ GYY4137 and ART treatment groups over time (*Figure 6—figure supplement 1F*). Moreover, we observed that prolonged treatment with GYY4137 in combination ART does not abrogate CD4$^+$ T-cell survival and proliferative capacity in the presence of anti-CD3/ anti-28 beads or Staphylococcal enterotoxin B (SEB) as T cell activation stimuli (*Figure 6—figure supplement 2A-D*). Additionally, GYY4137 treatment had no adverse effect on antigen (CMVpp65) specific CD4$^+$ and CD8$^+$ T cells activation and function upon ex vivo stimulation (*Figure 6—figure supplement 2E-J* and *Figure 6—figure supplement 3A-B*).

Altogether, using a range of cellular and immunological assays, we confirmed that the characteristics of an individual's viral reservoir remain preserved, and the suppression of viral RNA upon GYY4137 treatment is the result of H$_2$S-mediated inhibition of HIV-1 transcription rather than a reduction in proviral content or an altered CD4$^+$ T cell subsets. In sum, prolonged exposure to GYY4137 results in potent inhibition of viral reactivation, suggesting a new H$_2$S-based mechanism to neutralize bursts of virus reactivation under suppressive ART in vivo.

## GYY4137 prevents mitochondrial dysfunction in CD4$^+$ T cells of HIV-1 patients during viral rebound

Because virus reactivation upon depletion of CTH-mediated H$_2$S generation resulted in mitochondrial dysfunction and redox imbalance in U1, we tested if the elevation of H$_2$S levels by GYY4137 improves mitochondrial health and maintain redox homeostasis of primary CD4$^+$ T cells during virus reactivation ex vivo. As described earlier, CD4$^+$ T cells harboring latent virus upon prolonged (28 days) treatment of ART and ART+GYY4137 were stimulated by prostratin or left unstimulated and subjected to mitochondrial flux analysis. The unstimulated cells from both ART and ART+GYY4147 cultures did not show any difference in OCR (*Figure 7A*). In contrast, several features reflecting efficient mitochondrial activity such as basal respiration and ATP-coupled respiration were significantly higher in prostratin stimulated CD4$^+$ T cells in case of ART+GYY4137 treatment than ART alone (*Figure 7B–C*). Consistent with this, mitoROS generation upon stimulation with prostratin or other LRAs such as PMA/ionomycin was reduced in ART+GYY4137 treated CD4$^+$ T cells than ART alone (p=0.005 and p=0.014), and was nearly comparable to unstimulated cells (*Figure 7D–E*). The reduction in mitoROS could be a consequence of GYY4137-mediated increase in the expression of Nrf2-dependent antioxidant systems. We directly tested this by RT-qPCR analysis of a selected set of Nrf2-dependent genes on CD4$^+$ T cells treated with GYY4137 for 28 days. Consistent with our findings in U1, treatment with ART+ GYY4137 uniformly induced the expression of antioxidant genes in the latently infected CD4$^+$ T cells compared to cell treated with ART alone (*Figure 7F*).

Altogether, these data suggest that H$_2$S not only prevents virus reactivation but also improves mitochondrial bioenergetics and maintains redox homeostasis, which could be important for in vivo suppression of viral rebound and replenishment of the reservoir.

## Discussion

The major conclusion of our study is that HIV-1 reactivation is coupled to depletion of endogenous H$_2$S, which is associated with dysfunctional mitochondrial bioenergetics, in particular suppressed OXPHOS, GSH/GSSG imbalance, and elevated mitoROS. Decreased H$_2$S also impaired the expression of genes involved in the maintenance of redox balance, mitochondrial function, inflammation, and HIV-1 restriction, which correlates with reactivation of HIV-1. This conclusion is supported in multiple cell line models of HIV-1 latency. Chemical complementation with GYY4137 identified H$_2$S as an effector molecule. Finally, we confirmed the clinical relevance of our finding in primary CD4$^+$ T cells isolated from ART-suppressed patients to recapitulate features of HIV-1 latency and reactivation (*Kessing et al., 2017*). Overall, our data show that while H$_2$S deficiency reactivates HIV-1, the H$_2$S donor GYY4137 can potently inhibit residual levels of HIV-1 transcription during suppressive ART and block virus reactivation upon stimulation. Hence, our findings highlight that H$_2$S based therapies (*Szabó, 2007*) can be exploited to lock HIV-1 in a state of persistent latency by impairing the ability to reactivate.

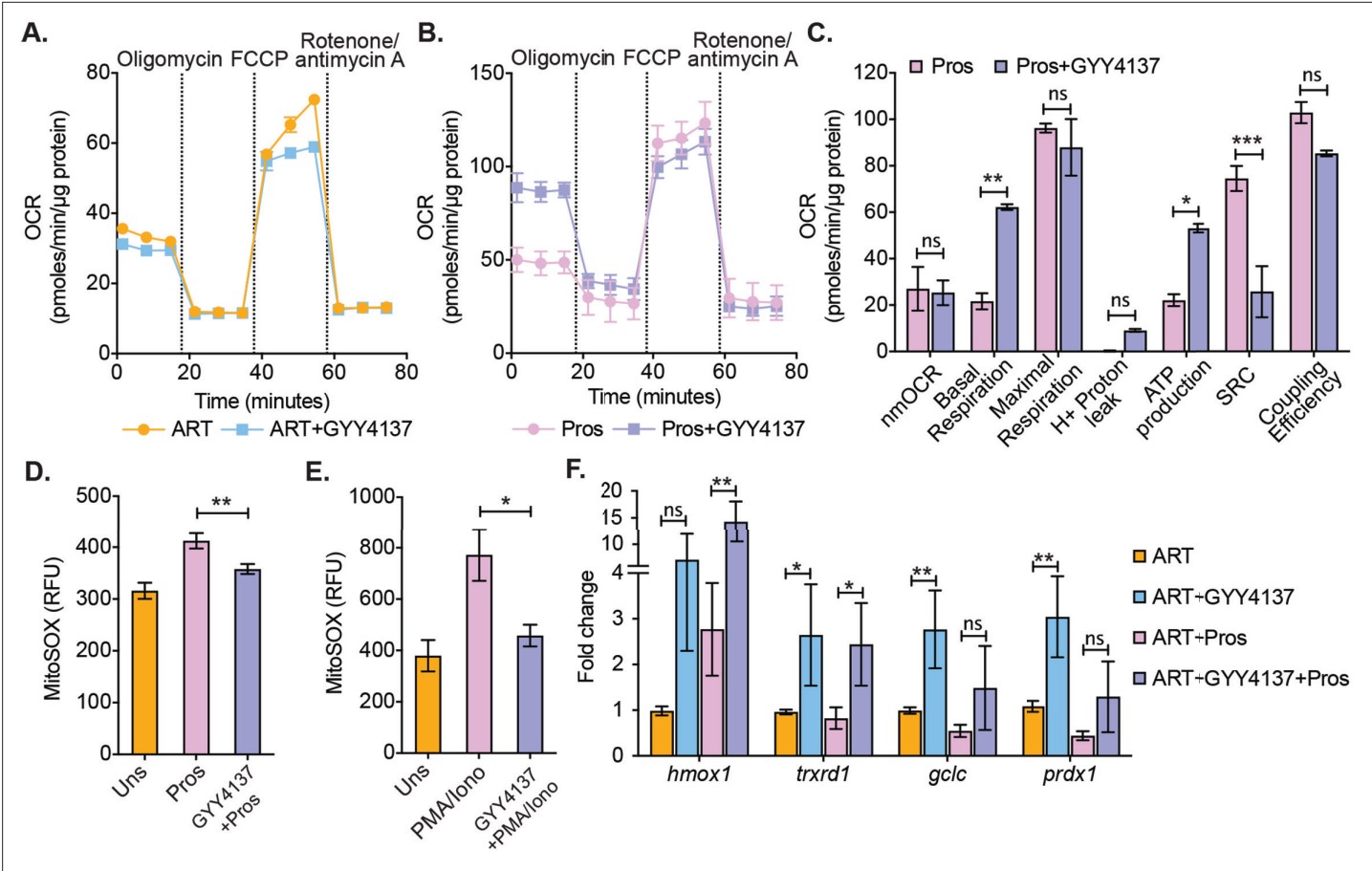

**Figure 7.** Effect of H₂S on mitochondrial respiration and ROS generation in latent CD4⁺ T cells derived from HIV-1 patients. (**A**) Primary human CD4⁺ T cells from HIV-1 infected subjects were activated and cultured ex vivo with ART or ART+GYY4137. On day 28, cells from ART or ART+GYY4137 treatment groups were harvested to assess mitochondrial respiration by using Seahorse XF mito-stress test as described in Materials and methods. (**B**) Cells from ART and ART+GYY4137 treated groups were stimulated with 1 µM prostratin 6 hr. Post-stimulation mitochondrial respiration profile was determined by Seahorse XF mito-stress test. (**C**) Various mitochondrial respiratory parameters derived from OCR measurement were determined by Wave desktop software. nmOCR; non-mitochondrial oxygen consumption rate and SRC; spare respiratory capacity. (**D**) On day 28, cells from both ART and ART+GYY4137 treatment groups were stimulated with 1 µM Prostratin for 6 hr. Cells were harvested and stained with 5 µM MitoSOX-Red dye for 30 min followed by washing. Samples were analyzed by flow cytometry. Unstimulated (Uns)- cells cultured under ART alone. (**E**) Both ART and ART+GYY4137 treated cells at day 14 post-activation were stimulated with 1 µg/ml PMA and 100 µg/ml ionomycin (Iono) for 6 h. Cells were harvested post-stimulation and stained with 5 µM MitoSOX-Red dye. Samples were analyzed by flow cytometry to assess mitoROS generation. Unstimulated (Uns)- cells cultured under ART alone. (**F**) CD4⁺ T cells from ART and ART+GYY4137 treated groups were stimulated with 1 µM prostratin on 28th day for 24 hr or left unstimulated. Cells were harvested to isolate total RNA and expression of *hmox1*, *txnrd1*, *gclc*, and *prdx1* were determined by RT-qPCR. Data obtained were normalized to internal control β₂ microglobulin (B2M). Error bar represents standard deviations from mean. Results are representative of data from three patient samples. *, p<0.05; **, p<0.01, by two-way ANOVA with Tukey's multiple comparison test. RT-qPCR, reverse transcription quantitative PCR.

The online version of this article includes the following figure supplement(s) for figure 7:

**Source data 1.** This file contains the source data used to make the graphs presented in *Figure 7*.

How does H₂S promote HIV-1 latency and suppress reactivation? Several studies revealed that HIV-1 reactivation is associated with loss of mitochondrial functions, oxidative stress, inflammation, and apoptosis (*Tyagi et al., 2020*; *Bhaskar et al., 2015*; *Khan et al., 2015*; *Badley et al., 2003*). Several of these biological dysfunctions are corrected by H₂S. For example, localized delivery of H₂S in physiological concentrations improves mitochondrial respiration, mitigates oxidative stress, and exerts anti-inflammatory functions (*Tinajero-Trejo et al., 2013*; *Kimura et al., 2010*; *Fu et al., 2012*; *Banerjee, 2011*; *Gao et al., 2012*; *Bhatia, 2012*; *Szabo et al., 2014*; *Szabó, 2007*). Our NanoString findings, XF flux analysis, and redox measurement support a mechanism whereby H₂S sufficiency promotes sustenance of mitochondrial health and redox homeostasis to control HIV-1 reactivation. Conversely, genetic depletion of endogenous H₂S promotes HIV-1 reactivation by decelerating OXPHOS, GSH/

GSSH imbalance, and elevated mitoROS. It is widely known that reduced OXPHOS leads to accumulation of NADH, which traps flavin mononucleotide (FMN) in the reduced state on respiratory complex I leading to mitoROS generation by incomplete reduction of $O_2$ (*Murphy, 2009*). Also, $H_2S$-depletion increased nmOCR, which indicates elevated activities of other ROS producing enzymes such as NADPH oxidase and cyclooxygenases/lipoxygenase (*Kim et al., 2008*; *Chacko et al., 2014*). All of these metabolic changes are likely contributors of overwhelming ROS and disruption of GSH/GSSG poise in U1-shCTH, resulting in HIV-1 reactivation. In fact, consistent with our findings, oxidative shift in GSH/GSSG poise, deregulated mitochondrial bioenergetics, and ROI has been reported to promote HIV-1 reactivation (*Tyagi et al., 2020*; *Bhaskar et al., 2015*). The addition of GYY4137 stimulated OXPHOS and blocked HIV-1 reactivation, which is in line with the known ability of $H_2S$ in sustaining respiration by acting as a substrate of cytochrome C oxidase (CcO) (*Szabo et al., 2014*). Also, $H_2S$ can prevent mitochondria-mediated apoptosis (*Hu et al., 2009*), which is crucial for HIV-1 reactivation and infection (*Khan et al., 2015*). Finally, activation of Nrf2-specific antioxidant pathways upon exposure to GYY4137 agrees with the potential of $H_2S$ in augmenting the cellular antioxidative defense machinery (*Pal et al., 2018*; *Kimura et al., 2010*; *Banerjee, 2011*). Importantly, cell lines (U1 and J1.1) modeling HIV-1 latency and PBMCs of individuals that naturally suppress HIV-1 reactivation (long-term non-progressors [LTNPs]) display robust capacities to resist oxidative stress and apoptosis (*Bhaskar et al., 2015*; *Cohen et al., 2004*).

Our transcription profiling showed a positive correlation between GYY4137-induced $H_2S$ sufficiency and HIV-1 restriction factors (e.g., YY1, APOBEC3, IRF1, and Nrf2) and negative correlation with HIV-1 trans-activators (e.g., NFKB and FOS), consistent with an interrelationship between cell metabolism and HIV-1 latency (*Valle-Casuso et al., 2019*; *Castellano et al., 2019*). Interestingly, a sustained induction of Nrf2-driven cellular antioxidant response is essential for the successful transition between productive and latent HIV-1 infection (*Bhaskar et al., 2015*; *Shytaj et al., 2020*). Moreover, inhibition of Nrf2 promoted viral transcription and increased ROS generation (*Shytaj et al., 2020*), whereas its activation reduced HIV-1 infection (*Furuya et al., 2016*). On this basis, we think that $H_2S$-mediated inhibition of HIV-1 is partly related to the activation of Nrf2-dependent antioxidant genes. In HIV-1 infection, the suppression of viral reactivation and pro-inflammatory mediator expression was paralleled by the inhibition of NF-κB. Interestingly, $H_2S$ inhibited activation of NF-κB in several models of inflammatory diseases, including hemorrhagic shock, lung injury, and paramyxovirus infections (*Cao et al., 2014*; *Li et al., 2015*). Further, recent studies have highlighted the anti-viral and anti-inflammatory role of $H_2S$ in SARS-CoV-2 infection and envisage the possibility of including $H_2S$ donors in COVID-19 therapy (*Renieris et al., 2020*; *Citi et al., 2020*; *Pozzi et al., 2021*). Similarly to these findings, we found that exposure of U1 with GYY4137 inhibited the NF-κB p65 subunit and reduced its binding to HIV-1 LTR. Currently, no antiretroviral drugs inhibit basal transcription of provirus and blocks reactivation upon therapy (ART) interruption. It has been suggested that epigenetic silencing will be an important prerequisite for persistent latency (*Lange et al., 2020*; *Kumar et al., 2015*). Our expression and ChIP-qPCR data suggest that $H_2S$ could exert its influence on HIV-1 replication machinery via activating YY1- an epigenetic silencer of HIV-1 LTR (*Coull et al., 2000*; *Bernhard et al., 2013*). Interestingly, another gasotransmitter NO modulates YY1 function via S-nitrosylation of its cysteine residues (*Bonavida and Baritaki, 2012*), indicating that YY1 is amenable to thiol-based-modifications and raises the possibility of YY1 S-persulfidation by $H_2S$ as a mechanism to epigenetically reconfigure HIV-1/promoter LTR.

The inefficiency of current ART in preventing virus rebound after therapy cessation poses a major hurdle to HIV-1 cure. Using a primary cell system that has been successfully used to recapitulate latency and reactivation seen in patients (*Kessing et al., 2017*), we confirmed that pretreatment with GYY4137 attenuated viral rebound in primary human $CD4^+$ T cells upon stimulation with prostratin. Collectively, our results suggest that $H_2S$ reduces HIV-1 transcriptional activity, promotes silencing of its promoter, and reduces its potential for reactivation. The virus-suppressing effects of $H_2S$ were observed alongside its beneficial consequences on cellular physiology (e.g., mitochondrial function and redox balance). Both mitochondrial dysfunction and oxidative stress are major complications associated with long-term exposure to ART and contribute to inflammation and organ damage (*Ibeh and Emeka-Nwabunnia, 2012*; *Hulgan and Gerschenson, 2012*; *Reyskens et al., 2013*). Future studies will investigate if the prolonged treatment with $H_2S$ donors results in a better outcome due to recruitment of specific repressor and/or epigenetic silencer of HIV-1 LTR without detrimental consequences

on cellular physiology. In this context, it is important to carefully calibrate the concentration of $H_2S$ donors. The higher concentration of $H_2S$ is associated with toxicity and side effects (*Shen et al., 2015*), whereas lower micro molar concentrations are beneficial (*Szabo et al., 2014*). The absorption and metabolism are reasonably well characterized for exogenously administered high doses of $H_2S$ (*Reiffenstein and Roth, 1992*; *Beauchamp et al., 1984*). The sulfide oxidation pathways are involved in the oxidation of excess $H_2S$ to thiosulfate and sulfate, which are cleared from the body via the urine in free or conjugated form (*Kage et al., 1997*). These observations will be important for exploring future possibilities to use $H_2S$ donors in clinical settings. Importantly, several $H_2S$ donors are currently being tested in phase 2 and phase 3 human clinical trials for other indications (*Wallace et al., 2020*; *Polhemus et al., 2015*).

Blocking HIV-1 rebound by inhibitors of viral factors (e.g., Tat) invariably results in the emergence of resistant mutants (*Mousseau et al., 2019*). In this context, blocking HIV-1 rebounds via H2S-directed modulation of host pathways could help in overcoming the problem of evolution of escape variants. In conclusion, we identify $H_2S$ as a central factor in HIV-1 reactivation. Our systematic mechanistic dissection of the role of $H_2S$ in cellular bioenergetics, redox metabolism, and latency unifies many previous phenomena associated with viral persistence.

## Materials and methods

**Key resources table**

| Reagent type (species) or resource | Designation | Source or reference | Identifiers | Additional information |
|---|---|---|---|---|
| Cell line (*H. sapiens*) | U1 | NIH HIV Reagent Program | Cat. #: ARP-165, RRID:CVCL_M769 | |
| Cell line (*H. sapiens*) | J1.1 | NIH HIV Reagent Program | Cat. #: ARP-1340, RRID:CVCL_8279 | |
| Cell line (*H. sapiens*) | ACH-2 | NIH HIV Reagent Program | Cat. #: ARP-349, RRID:CVCL_0138 | |
| Cell line (*H. sapiens*) | J-Lat 6.3 | NIH HIV Reagent Program | Cat. #: ARP-9846, RRID:CVCL_8280 | |
| Cell line (*H. sapiens*) | J-Lat 10.6 | NIH HIV Reagent Program | Cat. #: ARP-9849, RRID:CVCL_8281 | |
| Cell line (*H. sapiens*) | TZM-bl | NIH HIV Reagent Program | Cat. #: ARP-8129, RRID:CVCL_B478 | |
| Cell line (*H. sapiens*) | U937 | ATCC | RRID:CVCL_0007 | |
| Cell line (*H. sapiens*) | Jurkat | ATCC | RRID:CVCL_0367 | |
| Cell line (*H. sapiens*) | HEK293T | ATCC | RRID:CVCL_0063 | |
| Antibody | Anti-CBS (Rabbit monoclonal) | Abcam | Cat. #: ab140600, RRID:AB_2895036 | WB (1:1000) |
| Antibody | Anti-CTH (Rabbit polyclonal) | Abcam | Cat. #: ab151769, RRID:AB_2861405 | WB (1:1000) |
| Antibody | Anti-MPST (Rabbit polyclonal) | Abcam | Cat. #: ab154514, RRID:AB_2895038 | WB (1:1000) |
| Antibody | Anti-HIV-1 p24 (Mouse monoclonal) | Abcam | Cat. #: ab9071, RRID:AB_306981 | WB (1:1000) |
| Antibody | Anti-NRF2 (Rabbit monoclonal) | Cell Signaling Technology | Cat. #: 12721, RRID:AB_2715528 | WB (1:1000) |
| Antibody | Anti-KEAP1 (Rabbit polyclonal) | Cell Signaling Technology | Cat. #: 4678, RRID:AB_10548196 | WB (1:1000) |

*Continued on next page*

*Continued*

| Reagent type (species) or resource | Designation | Source or reference | Identifiers | Additional information |
|---|---|---|---|---|
| Antibody | Anti-NF-kB p65 (Mouse monoclonal) | Cell Signaling Technology | Cat. #: 6956, RRID:AB_10828935 | WB (1:1000), Chromatin IP (1:50) |
| Antibody | Anti-Phospho-NF-kB p65 (Ser536) (Rabbit monoclonal) | Cell Signaling Technology | Cat. #: 3033, RRID:AB_331284 | WB (1:1000) |
| Antibody | Anti-YY1 (Rabbit monoclonal) | Cell Signaling Technology | Cat. #: 63227, RRID:AB_2799641 | WB (1:1000), Chromatin IP (1:100) |
| Antibody | Anti-GAPDH (Mouse monoclonal) | Cell Signaling Technology | Cat. #: 97166, RRID:AB_2756824 | WB (1:1000), |
| Antibody | Anti-rabbit IgG, HRP-linked (Goat polyclonal) | Cell Signaling Technology | Cat. #: 7074, RRID:AB_2099233 | WB (1:10,000) |
| Antibody | Anti-mouse IgG, HRP-linked (Horse polyclonal) | Cell Signaling Technology | Cat. #: 7076, RRID:AB_330924 | WB (1:10,000) |
| Antibody | Anti-p24, KC57-RD1 (Mouse monoclonal) | Beckman Coulter | Cat. #: 6604667, RRID:AB_1575989 | Flow cytometry (1:100) |
| Recombinant DNA reagent | PLKO.1-puro non-mammalian shRNA control | Other | | Kind gift from Prof. D. K. Saini |
| Recombinant DNA reagent | psPAX2 | Other | | Kind gift from Prof. D. K. Saini |
| Recombinant DNA reagent | pMD2.G | Other | | Kind gift from Prof. D. K. Saini |
| Recombinant DNA reagent | pNL4-3 | NIH HIV Reagent Program | Cat. #: ARP-165 | |
| Peptide, recombinant protein | IL-2: PROLEUKIN | NOVARTIS | | |
| Commercial assay or kit | RNeasy mini kit | QIAGEN | Cat. #: 74106 | |
| Commercial assay or kit | Cysteine assay kit (fluorometric) | Sigma-Aldrich | Cat. #: MAK255 | |
| Commercial assay or kit | Glutathione assay kit | Cayman Chemical | Cat. #: 703002 | |
| Commercial assay or kit | SimpleChIP Enzymatic Chromatin IP Kit | Cell Signaling Technology | Cat. #: 9003 | |
| Commercial assay or kit | EasySep Human CD4$^+$ T Cell Isolation Kit | STEMCELL Technologies | Cat. #: 17952 | |
| Commercial assay or kit | CellTrace Violet Cell Proliferation Kit | Invitrogen | Cat. #: C34557 | |
| Commercial assay or kit | Live/Dead Fixable Aqua Dead Cell Stain Kit | Invitrogen | Cat. #: L34957 | |
| Chemical compound, drug | GYY4137 | Sigma-Aldrich | Cat. #: SML0100 | |
| Chemical compound, drug | PMA | Sigma-Aldrich | Cat. #: P8139 | |
| Chemical compound, drug | NAC | Sigma-Aldrich | Cat. #: A7250 | |

*Continued on next page*

*Continued*

| Reagent type (species) or resource | Designation | Source or reference | Identifiers | Additional information |
|---|---|---|---|---|
| Chemical compound, drug | PAG | Sigma-Aldrich | Cat. #: P7888 | |
| Chemical compound, drug | PHA | Thermo Fisher Scientific | Cat. #: R30852801 | |
| Chemical compound, drug | Efavirenz | NIH HIV Reagent Program | Cat. #: ARP-4624 | |
| Chemical compound, drug | Zidovudine | NIH HIV Reagent Program | Cat. #: ARP-3485 | |
| Chemical compound, drug | Raltegravir | NIH HIV Reagent Program | Cat. #: ARP-11680 | |
| Software, algorithm | GraphPad Prism | GraphPad Software (https://www.graphpad.com) | RRID:SCR_002798 | Version 9.0.0 for Macintosh |
| Software, algorithm | ImageJ | ImageJ (http://imagej.nih.gov/ij/) | RRID:SCR_003070 | |
| Software, algorithm | Wave Desktop | Agilent Technologies | RRID:SCR_014526 | Version 2.6 |
| Software, algorithm | nSolver | NanoString Technologies | RRID: SCR_003420 | Version 4.0 |
| Software, algorithm | FlowJo | (https://www.flowjo.com/solutions/flowjo) | RRID: SCR_008520 | 9.9.6 and v10 |

## Study design

The primary objective of this study was to understand the role $H_2S$ gas in modulating HIV-1 latency and reactivation. First, we examined the differential expression of enzymes involved in $H_2S$ biogenesis and levels of $H_2S$ upon HIV-1 latency and reactivation. Next, we genetically silenced the expression of the main $H_2S$ producing gene CTH and showed its importance in maintaining HIV-1 latency. We performed detailed mechanistic studies on understanding the role of CTH in regulating cellular antioxidant response, mitochondrial respiration, ROS generation to maintain HIV-1 latency. Finally, pharmacological donor of $H_2S$ (GYY4137) was used to reliably increase endogenous $H_2S$ levels and to study its consequence on HIV-1 latency and rebound. Primary $CD4^+$ T cells derived from ART-treated HIV-1 infected patients were used to assess the effect of GYY4137 in modulating HIV-1 rebound by improving mitochondrial bioenergetics and mitigating redox stress.

## Mammalian cell lines and culture conditions

The human pro-monocytic cell line U937, $CD4^+$ T lymphocytic cell line Jurkat, HEK293T were procured from ATCC, Manassas, VA. The chronically infected U1, J1.1, ACH-2, J-Lat 6.3, J-Lat 10.6, and TZM-bl cell lines were obtained through NIH HIV Reagent Program, Division of AIDS, NIAID, NIH, USA. All cell lines are verified to be mycoplasma free by EZdetect PCR Kit for Mycoplasma Detection (HIMEDIA). The cell lines were cultured in RPMI 1640 (Cell Clone) supplemented with L-glutamine (2 mM), 10% fetal bovine serum (FBS; MP Biomedicals), penicillin (100 units/ml), streptomycin (100 μg/ml) at 37°C and 5% $CO_2$. HEK293T and TZM-bl cells were cultured in Dulbecco's modified Eagle's medium (Cell Clone) supplemented with 10% FBS.

## Chemical reagents

Sodium hydrosulfide (NaHS), morpholin-4-ium 4-methoxphenyl(morpholino) phosphinodithioate dichloromethane complex (GYY4137), Phorbol-12-myristate-13-acetate (PMA), N-acetyl cysteine (NAC), Prostratin, DL-Propargylglycine (PAG), and L-Cysteine were purchased from Sigma-Aldrich. Recombinant human TNF-α was purchased from InvivoGen. The antiretroviral drugs efavirenz, zidovudine, raltegravir, and lamivudine were obtained through the NIH HIV Reagent Program.

## Latent viral reactivation

Latently infected U1, J1.1, and J-Lat 10.6 ($2×10^5$ cells/ml) were stimulated with PMA (5 ng/ml) or TNF-α (100 ng/ml) for the time indicated in the figure legends. HIV-1 reactivation was determined by intracellular *gag* RT-qPCR or p24 estimation in supernatant by HIV-1 p24 ELISA (J. Mitra and Co. Pvt. Ltd., India). J-Lat 6.3 cells were stimulated with Prostratin (2.5 µM) for 24 hr and HIV-1 reactivation was assessed by estimating GFP+ cells (excitation: 488 nm; emission: 510 nm) using BD FACSVerse flow cytometer (BD Biosciences). The data were analyzed using FACSuite software (BD Biosciences).

## Reverse transcription quantitative PCR

Total cellular RNA was isolated by RNeasy Mini Kit (QIAGEN), according to the manufacturer's protocol. RNA (500 ng) was reverse transcribed to cDNA (iScript cDNA Synthesis Kit, Bio-Rad), subjected to quantitative real-time PCR (iQ SYBR Green Supermix, Bio-Rad), and performed using the Bio-Rad C1000 real-time PCR system. HIV-1 reactivation was assessed using gene-specific primers (*Supplementary file 1f*). The expression level of each gene is normalized to human β-actin as an internal reference gene.

## Western blot analysis

Total cell lysates of PMA-treated and untreated U1, J1.1, U937, and Jurkat cell lines were prepared using radioimmunoprecipitation (RIPA) lysis buffer (50 mM Tris [pH 8.0], 150 mM NaCl, 1% Triton X-100, 1% sodium deoxycholate, 0.1% SDS [sodium dodecyl sulfate], 1× protease inhibitor cocktail (Sigma-Aldrich), 1× phosphatase inhibitor cocktail [Sigma-Aldrich]). After incubation on ice for 20 min, the lysates were centrifuged at 12,000 rpm, 4°C for 15 min. Clarified supernatant was taken and total protein concentration was determined by Bicinchoninic Acid Assay (Pierce, Thermo Fisher Scientific). Total protein extracts were separated by SDS-PAGE and transferred onto polyvinylidene difluoride membranes. Membranes were probed with an anti-CBS (EPR8579), CTH (ab151769), MPST (ab154514), and anti-HIV-1 p24 (ab9071) from Abcam; Nrf2 (CST-12721), Keap1 (CST-4678), NF-κB p65 (CST-6956), phospho-NF-κB p65 (Ser536) (CST-3033), YY1 (CST-63227), and GAPDH (CST-97166) from Cell Signaling Technologies, Inc; and anti-rabbit IgG (CST-7074) and anti-mouse IgG (CST-7076) were used a secondary antibody. Proteins were detected by ECL and visualized by chemiluminescence (PerkinElmer, Waltham, MA) using the Bio-Rad Chemidoc Imaging system (Hercules, CA). For membrane reprobing, stripping buffer was used (2% SDS [w/v], 62 mM Tris-Cl buffer [0.5 M, pH 6.7] and 100 mM β-mercaptoethanol) for 20 min at 55°C. After extensive washing with phosphate-buffered saline (PBS) containing 0.1% Tween 20 (Sigma-Aldrich), membrane was blocked and reincubated with desired antibodies.

## H₂S detection assays

Endogenous $H_2S$ levels of U1 and U937 cells were detected using $H_2S$ specific fluorescent probe as described (*Chen et al., 2013*). Briefly, cells were treated with PMA (5 ng/ml) for 24 hr, washed with 1× PBS, and stained with 200 µM 7-Azido-4-Methylcoumarin (AzMC) (Sigma-Aldrich). The stained cells were mounted on a glass slide and visualized using Leica TCS SP5 confocal microscope (excitation: 405 nm; emission: 450 nm). Images obtained were analyzed using LAS AF Lite software (Leica Microsystems) and semi-quantification of 50 cells was performed using ImageJ software.

$H_2S$ generation was also measured using methylene blue assay. The supernatant of U1 cells treated with NaHS or GYY4137 was incubated with Zinc acetate (1%) and NaOH (3%) (1:1 ratio) to trap $H_2S$ for 30 min. The reaction was terminated using 10% trichloroacetic acid solution. Following this, reactants were incubated with 20 mM *N*,*N*-dimethylphenylendiamine (NNDPD) in 7.2 M HCl and 30 mM $FeCl_3$ in 1.2 M HCl for 30 min and absorbance was measured at 670 nm. The concentration of $H_2S$ was determined by plotting absorbance on a standard curve generated using NaHS (0–400 µM; $R^2$=0.9982).

## Stable cell line generation

For generating CBS and CTH knockdown in U1 and J1.1 cells, we used validated pooled gene-specific shRNAs from the RNAi Consortium (TRC) library (Sigma-Aldrich, USA; shRNA sequences given in *Supplementary file 1a*). The lentiviral particles were generated in HEK293T cells using the packaging vectors, psPAX2 and pMD2.G. The pLKO.1-puro vector encoding a non-mammalian targeting shRNA (shNT) was used as a control. The U1 cells were transduced with lentiviral particles in opti-MEM

containing polybrene (10 µg/ml) for 6 hr. Cells were washed and stable clones were selected in culture medium containing 250 ng/ml of puromycin. Total RNA or cell lysates were prepared to validate knockdown of CBS and CTH.

## Intracellular HIV-1 P24 staining

For intracellular p24 staining, U1-shCTH and U1-shNT cells were stimulated with PMA (5 ng/ml), washed with PBS followed by fixation and permeabilization using a fixation and permeabilization kit (eBiosciences). Permeabilized cells were then incubated with 50 µl of 1:100 dilution of phycoerythrin (PE)-conjugated mouse anti-p24 monoclonal antibody (KC57-RD1; Beckman Coulter, Inc) for 30 min at room temperature. After incubation, the cells were washed two times and the fluorescence of stained samples was acquired using BD FACSVerse flow cytometer (BD Biosciences). The data were analyzed using FACSuite software (BD Biosciences).

## Cysteine estimation

The quantification of cysteine was determined using Cysteine assay kit (Sigma-Aldrich). U1-shNT and U1-shCTH cells ($10^7$ cells) were harvested and lysed by sonication in assay buffer on ice. Lysates were clarified by centrifugation at 12,000 rpm, 4°C for 10 min. Then, 10 µl of sample was added to each well and incubated with enzyme mix and detection reagent as per the manufacturer's instruction. The assay plate was read using fluorescence plate reader at Ex/Em=365/450 nm in kinetic mode at room temperature. The concentration of cysteine in sample was calculated from standard curve.

## Cysteine deprivation and HIV-1 reactivation

U1 cells ($2 \times 10^5$ cells/ml) were cultured in complete RPMI 1640 medium containing 208.16 µM L-cystine (as cysteine source) and 100 µM L-methionine (denoted as [+Cys]) or RPMI 1640 medium without L-methionine and L-cystine (Sigma-Aldrich, Cat #7513) supplemented with 100 µM L-methionine (Sigma-Aldrich, Cat #64319) (denoted as cysteine-free RPMI [−Cys]), 10% dialyzed FBS (Gibco), penicillin (100 units/ml), and streptomycin (100 µg/ml). Cells were harvested at indicated time points and viral reactivation was analyzed by *gag* RT-qPCR.

## NanoString nCounter assay

Total RNA was isolated using a RNeasy Mini Kit (QIAGEN) according to the manufacturer's instructions. RNA concentration and purity were measured using a Nanodrop spectrophotometer (Thermo Fisher Scientific, Waltham, MA), bioanalyzers systems (Agilent Technologies, Inc), and Qubit assays (Thermo Fisher Scientific). An nCounter gene expression assay was performed according to the manufacturer's protocol. The assay utilized a custom-made NanoString codeset designed to measure 185 genes, including six house-keeping genes (*Supplementary file 1b*). This custom-made panel included genes associated with oxidative stress and HIV-1 infection (*Supplementary file 1b*). All genes were assayed simultaneously in multiplexed reactions and analyzed by fully automated nCounter Prep Station and digital analyzer (NanoString Technologies). The data were normalized to B2M, used as a housekeeping gene due to its minimum % CV across the samples, and analysis was done using nSolver 4.0 software.

## Measurement of oxygen consumption rates

OCRs were measured using a Seahorse XFp extracellular flux analyzer (Agilent Technologies) as per the manufacturer's instructions. Briefly, cells (U1 or Primary CD4+ T cells) were seeded at a density of $10^4$–$10^5$ per well in a Seahorse flux analyzer plate precoated with Cell-Tak (Corning). Cells were incubated for 1 hr in a non-CO₂ incubator at 37°C before loading the plate in the seahorse analyzer. To assess mitochondrial respiration, three OCR measurements were performed without any inhibitor in XF assay media to measure basal respiration, followed by sequential addition of oligomycin (1 µM), an ATP synthase inhibitor (complex V), and three OCR measurements to determine ATP-linked OCR and proton leakage. Next, cyanide-4-(trifluoromethoxy)phenylhydrazone (FCCP; 0.25 µM), was injected to determine the maximal respiration rate and the SRC. Finally, rotenone (0.5 µM) and antimycin A (0.5 µM), inhibitors of NADH dehydrogenase (complex I) and cytochrome *c* - oxidoreductase (complex III), respectively, were injected to completely shut down the ETC to analyze nmOCR. Seahorse data

were normalized to total amount of protein (μg) and mitochondrial respiration parameters were analyzed using Wave Desktop 2.6 software (Agilent Technologies).

## Estimation of intracellular glutathione content and ROS

Total cell lysate was prepared from $10^7$ cells using sonication in MES (2-(N-morpholino) ethanesulfonic acid) buffer. Lysates were clarified by centrifugation and total protein concentration was estimated using BCA assay. Total glutathione, reduced glutathione (GSH), and oxidized glutathione (GSSG) were measured using the glutathione assay kit (Cayman Chemical, Ann Arbor, MI) according to the manufacturer's instructions. To measure ROS, cells were loaded with 10 μM of CM-H$_2$DCFDA (excitation: 492 nm; emission; 517 nm) or 5 μM of MitoSOX Red (excitation: 510 nm; emission; 580 nm) for 30 min at 37°C and exposed to H$_2$O$_2$ (100 μM) or antimycin A (2 μM) or PMA (5 ng/ml) or left untreated at 37°C and 5% CO$_2$. The fluorescence of stained samples was acquired using BD FACSVerse flow cytometer (BD Biosciences). Data were analyzed using FlowJo software (BD Biosciences).

## Virus production

HIV-1 particle production was carried out using Lipofectamine 2000 transfection reagent (Invitrogen, Life Technologies), according to the manufacturer's protocol, in HEK293T cells using HIV-1 NL4-3 DNA (NIH HIV Reagent Program, Division of AIDS, NIAID, NIH). The medium was replaced with fresh medium at 6 hr post-transfection, and supernatants were collected after 60 hr, centrifuged (10 min, 200×g, room temperature), and filtered through a 0.45-μm-pore-size membrane filter (MDI; Membrane Technologies) to clear cell debris. Virus was concentrated using 5× PEG-*it* (System Biosciences) as per the manufacturer's protocol and virus pellet obtained was aliquoted in opti-MEM and stored at –80°C. Viral titration was done using HIV-1 reporter cell line, TZM-bl (NIH HIV Reagent Program) as described earlier (*Cherne et al., 2019*).

## Chromatin immunoprecipitation and quantitative genomic PCR

The chromatin immunoprecipitation (ChIP) assays were performed using SimpleChIP Enzymatic Chromatin IP Kit (Magnetic Beads) (Cell Signaling Technologies, Inc, MA) according to the manufacturer's instructions. Briefly, $10^7$ U1 cells were pre-treated with GYY4137 (5 mM) for 6 hr or left untreated and then stimulated with PMA (30 ng/ml) for 4 hr. Cells were fixed using 1% formaldehyde, neutralized with glycine, and harvested in ice-cold PBS. Cells were lysed and nuclei were sheared by sonication using Bioruptor Pico (Diagenode Inc) to obtain DNA fragments of 200–500 nucleotides. The clarified lysates were immunoprecipitated using ChIP-grade anti-NF-κB p65 (CST-6956) or anti-YY1 (CST-63227), or normal rabbit IgG (as a negative control) for overnight at 4°C followed by incubation with ChIP-grade protein G magnetic beads at 4°C for 4 hr. Chromatin was eluted from protein G beads, reverse-cross linked, and DNA was column purified. Quantitative genomic PCR was done by SYBR green-based real-time PCR using primers spanning the ER, RBE I, and RBE III regions on HIV-1 LTR. The GAPDH gene was used as the reference gene to see non-specific binding. Total input (10%) was used to normalize equal amount of chromatin taken across the samples. The relative proportion of co-immunoprecipitated DNA fragments were determined with the help of threshold cycle (C$_T$) values for each qPCR product using the equation ($100 \times 2^{[CT(input-3.32)- CT(IP)]}$). The data obtained were represented as fold enrichment normalized to IgG background for each IP reaction.

## Subject samples

PBMCs were collected from five HIV-1 seropositive subjects on stable suppressive ART and three healthy HIV-seronegative donors (*Supplementary file 1e*). All subject signed informed consent forms approved by the Indian Institute of Science, Bangalore and Bangalore Medical College and Research Institute (BMCRI) review boards (Institute human ethics committee [IHEC] No-3-14012020). The PBMCs isolated from blood samples using Histopaque-1077 (Sigma-Aldrich) density gradient centrifugation was used for CD4$^+$ T cells purification. The CD4$^+$ T cells were purified from 50×10$^6$ PBMCs using EasySep Human CD4$^+$ T cell isolation kit (STEMCELL Technologies).

## Infection of primary CD4$^+$ T cells

Primary CD4$^+$ T cells were cultured for 3 days after isolation in RPMI 1640 supplemented with 10% FBS, 100 U/ml interleukin-2 (IL-2; PROLEUKIN; NOVARTIS), and activated by adding the Dynabeads

Human T-Activator CD3/CD28 using a bead/cell ratio of 1:2. Post-activation CD4$^+$ T cells were pre-treated with 800 µM GYY4137 for 6 hr, and infected with HIV-NL4.3 (1 ng p24/ 10$^6$ cells) by spinoculation at 1000×$g$ for 90 min at 32°C. Following infection, cells were washed and cultured in the presence or absence of 800 µM GYY4137 in complete media containing 100 U/ml IL-2. To quantify the virion release, supernatant was harvested from infected cells and centrifuged at 400×$g$ for 10 min and virus concentration was estimated by HIV-1 p24 ELISA (J. Mitra and Co. Pvt. Ltd., India) according to the manufacturer's instruction.

### Generation of expanded primary CD4$^+$ T cells from aviremic subjects

Primary CD4$^+$ T cells from five ART suppressed, aviremic HIV-1 infected donors were cultured at 37°C in a 5% (v/v) CO$_2$ humidified atmosphere in RPMI 1640 medium (Gibco) with GlutaMAX, HEPES, 100 U/ml IL-2, 1 µg/ml Remel phytohemagglutinin (PHA) (Thermo Fisher Scientific), gamma-irradiated feeder PBMCs (healthy control) and either ART alone (100 nM efavirenz, 180 nM zidovudine, and 200 nM raltegravir) or ART +100 µM GYY4137. After 7 days, CD4$^+$ T cells were cultured only with ART alone or ART+GYY4137 and 100 U/ml IL-2. For stimulation experiments on day 28, ART and ART+GYY4137 were washed off and 1×10$^6$ cells were treated with 1 µM prostratin for 24 hr.

### HIV-1 RNA and DNA isolation from primary CD4$^+$ T cells

CD4$^+$ T cells (1×10$^6$) from ART or ART+GYY4137 treated groups were harvested to isolate total RNA using Qiagen RNAeasy isolation kit and 200 ng of total RNA was reverse transcribed (iScript cDNA synthesis kit, Bio-Rad). Reverse transcribed cDNA was diluted tenfold and amplified using primers against HIV-1 LTRs and seminested—PCR was performed using primers and probe listed in *Supplementary file 1f*. Serially diluted pNL4.3 plasmid was used to obtain the standard curve. Isolation and RT-qPCR of total HIV-1 DNA were performed as described earlier (*Kessing et al., 2017*). Briefly, 1×10$^6$ cells were lysed (10 mM Tris-HCl, 50 nM KCl, 400 mg/ml proteinase K) at 55°C for 16 hr followed by inactivation at 95°C for 5 min. Digested product was used as a template to set up the first PCR with Taq polymerase (NEB), 1× Taq buffer, dNTPs, HIV-1 and CD3 primers for 12 cycles. The second-round amplification was done using seminested PCR strategy wherein tenfold dilution of first round PCR product was used as a template, HIV-1, and CD3 primers/probes, Taqman Fast Advance master mix (Applied Biosystems) using SetupOnePlus Real-time PCR system (Applied Biosystems). DNA isolated from ACH2 cells that contain a single copy of HIV-1 per cell was used to obtain standard curve.

### Surface marker analysis of primary CD4$^+$ T cells derived from HIV-1 patients

CD4$^+$ T cells derived from HIV-1 infected patients were stained for surface markers using monoclonal antibodies: CD4-BUV395 (SK3), CD45RA-APC-H7 (HI100), CD27-BV785 (O323), CCR7-Alexa 647 (G043H7), CD38-PE-Cy5 (HIT2), and CD127-PerCP-Cy5.5 (eBioRDR5). Additionally, cells were stained with Live/Dead fixable Aqua dead cell stain or AviD (Invitrogen) to exclude dead cells from the analysis as per the manufacturer's instructions. PBMCs from HIV positiveART-treated patients were incubated with ART or ART in combination with 100 µM GYY4137 and either left unstimulated or stimulated with either CMVpp65 (JPT Technologies) or anti-CD3/anti-CD28 beads (at a bead:cell ratio of 0.25:1) for 18 hr. Cells were stained with antibodies for analysis of cell surface markers CD3-BV570 (UCHT1), CD4-FITC (RPA-T4), CD8-PerCP-Cy5.5 (SK1), CD25-BV421 (M-A251), CD69-PE (L78), and intracellular cytokines IFNγ-Alexa Fluor 700 (B27) and IL-2-APC (MQ1-17H12). Stained samples were run on BD FACSAria Fusion flow cytometer (BD Biosciences, San Jose, CA) and data were analyzed with FlowJo version 9.9.6 and v10 (Treestar, Ashland, OR).

### Cell proliferation assay

PHA/IL-2 expanded CD4$^+$ T cell lines cultured for 14 days in the presence of ART or ART in combination with 100 µM GYY4137 were washed and rested overnight. Cells were labeled with 0.5 µM cell trace violet (CTV) (Invitrogen) and incubated at 37°C for 7 min in the dark, after which CTV uptake was stopped by addition of 5 ml ice-cold complete RPMI media. Cells were pelleted down washed once with complete RPMI and used in the cell proliferation assay. CTV-labeled cells were seeded at a density of 0.5×10$^6$/well and stimulated with either 1.7 µg/ml CMVpp65 (JPT Technologies), anti-CD3/anti-CD28 T cell activator beads at a bead:cell ratio of 0.25:1 (Thermo Fisher Scientific) or 100 ng/ml

SEB (Sigma-Aldrich). After 3 days, IL-2 (20 U/ml) supplemented medium was added to the cells and cultured for another 2 days. Proliferation was measured by CTV dilution after 5 days by flow cytometry. In some experiments, after 5 days of culture CTV-labeled cells were pelleted and stained with antibodies to CD3, CD4, and CD25 and CD25 expression was analyzed by flow cytometry.

## Statistical analysis

All statistical analyses were performed using GraphPad Prism software for Macintosh (version 9.0.0). The data values are indicated as mean ± S.D. For statistical analysis Student's $t$-test (in which two groups are compared) and one-way or two-way ANOVA (for analysis involving multiple groups) were used. For flow cytometry experiments, data are represented as means ± SEM and p-value between paired samples was determined by Wilcoxon matched-pairs signed-rank test. Analysis of NanoString data was performed using the nSolver platform. Differences in p-values <0.05 and fold change >1.5 were considered significant.

## Acknowledgements

The authors are grateful to Prof. C Grundner at the Seattle Children's Research Institute for critical reading of the manuscript and valuable input. The authors acknowledge Prof. D K Saini at Department of Molecular Reproduction, Development and Genetics, IISc for providing shRNA constructs used in this study. The authors are grateful to Ms. S Kumar and Dr. N R Sundaresan at Department of Microbiology and Cell Biology, IISc for helping with ChIP experiments carried out in this study. The authors gratefully acknowledge the NanoString services provided by TheraCUES Innovations Pvt Ltd, Bangalore. This work was supported by Wellcome Trust-Department of Biotechnology (DBT) India Alliance grant IA/S/16/2/502700 (AS) and in part by DBT grants BT/PR13522/COE/34/27/2015, BT/PR29098/Med/29/1324/2018, and BT/HRD/NBA/39/07/2018–19 (AS), DBT-IISc Partnership Program grant 22-0905-0006-05-987 436, Infosys Foundation, and DST-FIST. AS is a senior fellow of Wellcome Trust-DBT India Alliance. VKP is grateful to Indian Institute of Science for fellowship.

## Additional information

### Funding

| Funder | Grant reference number | Author |
|---|---|---|
| The Wellcome Trust DBT India Alliance | IA/S/16/2/502700 | Amit Singh |
| Department of Biotechnology, Ministry of Science and Technology, India | BT/PR13522/COE/34/27/2015 BT/PR29098/Med/29/1324/2018 and BT/HRD/NBA/39/07/2018-19 | Amit Singh |
| Department of Biotechnology, Ministry of Science and Technology, India | 22-0905-0006-05-987 436 | Amit Singh |
| Infosys Foundation | CIDR | Amit Singh |
| Department of Science and Technology | DST-FIST | Amit Singh |

The funders had no role in study design, data collection and interpretation, or the decision to submit the work for publication.

### Author contributions

Virender Kumar Pal, Conceptualization, Formal analysis, Investigation, Methodology, Writing - original draft, Writing – review and editing; Ragini Agrawal, Formal analysis, Investigation; Srabanti Rakshit, Investigation, Methodology; Pooja Shekar, Diwakar Tumkur Narasimha Murthy, Investigation, Resources; Annapurna Vyakarnam, Formal analysis, Investigation, Methodology, Resources, Writing

– review and editing; Amit Singh, Conceptualization, Formal analysis, Funding acquisition, Investigation, Project administration, Supervision, Writing - original draft, Writing – review and editing

### Author ORCIDs
Virender Kumar Pal ![ORCID] http://orcid.org/0000-0001-8378-1823
Amit Singh ![ORCID] http://orcid.org/0000-0001-6761-1664

### Ethics
Peripheral blood mononuclear cells (PBMCs) were collected from five aviremic HIV-seropositive subjects on stable suppressive ART. All subjects provided signed informed consent approved by the Indian Institute of Science, and Bangalore Medical College and Research Institute review boards (IHEC No.- 3-14012020).

### Decision letter and Author response
Decision letter https://doi.org/10.7554/eLife.68487.sa1

## Additional files

### Supplementary files
• Supplementary file 1. Supplementary materials associated with this article. (a) Sequences of shRNA clones. (b). List of genes used for NanoString nCounter Gene Expression Analysis. (c). List of differentially expressed genes from NanoString nCounter Gene Expression analysis for U1-shNT, U1-shCTH, U1-shNT+ PMA and U1-shCTH+ PMA samples. (d). List of differentially expressed genes from NanoString nCounter Gene Expression analysis for U1 untreated, GYY4137, PMA and GYY4137+ PMA treated samples. (e). Characteristic of ART treated HIV-1 infected study participants. (f). List of primers and probes used in this study.

• Transparent reporting form

### Data availability
All data generated or analysed during this study are included in the manuscript and supporting files. Source data files have been provided for Figures 2 and 5.

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
