## [Editor Report]

Pal and colleagues show that hydrogen sulfide (H_2_S) inhibits HIV replication and reactivation by a variety of mechanisms including inhibition of NF-κB and recruitment of the epigenetic silencer, YY1, to the HIV promoter. They further report that H_2_S helps in maintaining mitochondrial bioenergetics and redox homeostasis. Altogether, the study provides novel insights into the mechanisms underlying HIV transcription and reactivation.

---

## [Decision Letter]

**Decision letter after peer review:**

[Editors’ note: the authors submitted for reconsideration following the decision after peer review. What follows is the decision letter after the first round of review.]

Thank you for submitting your work entitled "Hydrogen sulfide blocks HIV rebound by maintaining mitochondrial bioenergetics and redox homeostasis" for consideration by *eLife*. Your article has been reviewed by 2 peer reviewers, including Frank Kirchhoff as the Reviewing Editor and Reviewer #1, and the evaluation has been overseen by a Senior Editor.

We are sorry to say that, after consultation with the reviewers, we have decided that your work will not be considered further for publication by *eLife* in the present form.

We feel that your manuscript is well performed and contains interesting data from a basic research perspective. However, as an inhibitor of HIV reactivation, H2S donors would provide clinical utility if these are more effective and safer than ART. Adding an H2S donor during ART does not really seem to be particularly useful due to viral rebound after treatment interruption. It seems that GYY4137 works essentially as a NFkB inhibitor to suppress reactivation of HIV, which also raises concerns about side-effects. Altogether, we found your findings interesting but better suitable for a more specialized journal.

*Reviewer #1:*

Hydrogen sulfide (H2S) is best known for it smell of rotten eggs and its high toxicity due to interference with oxygen transport. Recent studies suggest, however, that H2S is involved in various physiological and pathological processes and might exert beneficial therapeutic effects at very low doses. Thus, H2S-releasing compounds are considered for the treatment of e.g. cardiovascular diseases. In the present study, Pal et al. report that H2S also plays an important role in reactivation of HIV from latency. The first show that HIV reactivation reduces the levels of endogenous H2S in cell lines. They further show that suppression of the H2S producing enzyme CTH enhances HIV reactivation and modulates the expression of cellular genes associated with apoptosis and mitochondrial function. The authors also provide evidence that endogenous production of H2S or release from small molecule H2S donors allows U1 and Jurkat cells to maintain normal redox homeostasis and mitochondrial bioenergetics and suggest that this promotes HIV-1 latency. Finally, they provide evidence that the H2S donor GYY4137 modulates Nrf2, NF-κB, and YY1 pathways and suppresses HIV reactivation in latently infected T cells from HIV-infected individuals. The study addresses an important topic from a different angle. It is comprehensive and provides insights into the mechanisms underlying the effect of H2S on HIV latency. In addition, the impact of H2S donors on reactivation of HIV is of significant interest. Limitations of the study are that potential risks and problems associated with H2S as therapeutic agent are not addressed. H2S is highly toxic; thus, dosage will be a major challenge. Treatment would not only affect cells harboring latent HIV but all cells and the H2S-mediated mechanisms proposed to suppress HIV reactivation, such as inhibition of NF-κB and altered metabolism, would be expected to cause side-effects. Finally, the authors propose to combine H2S donors with ART to suppress HIV reactivation. However, effective ART usually prevents viral replication with high efficiency. Thus, rebound after treatment interruption (and not under ART) is the problem. In the end, a functional cure only makes sense if it doesn't need to be combined with ART and does not require daily treatment.

1. Problems of dosing and side-effects of H2S donors should be critically discussed. The advantage of combining an H2S donor with ART did not become clear to me since the latter very reliably prevents viral replication.

2. The study is comprehensive and it is appreciated that the authors analyzed the impact of an H2S donor in primary CD4^+^ T cells. However, all mechanistic data were obtained in cell lines that may not faithfully represent the status of latently HIV infected cells in vivo. Ideally, some key finding should be confirmed in primary cells. Otherwise this limitation should be mentioned. Similarly, the cell line used harbor copies of T cell line adapted HIV-1 strains that may differ from proviruses representing the latent HIV reservoirs in vivo.

3. The authors propose to repurpose H2S donors for clinical treatment. Are such agents already used in the clinic or currently just examined in preclinical animal models?

4. The authors state that H2S "bolsters the anti-HIV potential of immune cells". This conclusion does not seem to be fully warranted based on the data presented. Thus, the statement should be cautioned or further substantiated.

*Reviewer #2:*

While the idea of H2S acting as a regulator of HIV-1 latency is interesting, there is a long list of inconsistencies that hamper the enthusiasm for this manuscript.

Much of the cited literature that is used to make the case for their hypothesis is very old and actually refers to active HIV infection and patient studies prior to ART. Also, the literature they cite regarding the role of H2S as an antimicrobial agent seem to be limited to tuberculosis infection.

The choice of the latently infected model cell lines is rather unfortunate. There are much better defined models out there these days than J1.1 or U1 cells, such as the J-LAT cells from the Verdin lab or the various reporter cell lines generated by Levy and co-workers. In particularly, U1 cells should not be considered as latently infected, as the virus has a defect in the Tat/TAR axis and is mostly just transcriptionally attenuated. It is unclear why the authors only use J-LAT cells for one of the last experiments.

It is further unclear why the authors perform most of the experiments using U1 cells, which are considered promonocytic, but in the end seek to demonstrate the influence of H2S on latent HIV-1 infection in CD4 T cells. Performing all experiments in J1.1 or better J-LAT cells would have seemed more intuitive.

The authors suggest that H2S production would control latent HIV-1 infection and reactivation. Regarding the idea that CBS, CTH or possibly MPST would control latent infection as a function of their ability to produce H2S from different sources, there are several questions. First, if H2S is the primary factor, why would the presence of e.g. MPST no compensate for the reduction of CTH? Second, why would J1.1 and U1 cells both host latent HIV-1 infection events, however, their CDB/CTH/MPST composition is completely different? Third, natural variations in CTH expression caused by culture over time are larger than variations caused by PMA activation.

Also, the statement that H2S production as exerted per loss of CTH would control reactivation is not supported by the kinetic data. In latently HIV-1 infected T cell lines or monocytic cell lines, PMA-mediated HIV-1 reactivation at the protein level is usually almost complete after 24 hours, but at this time point the difference between e.g. CTH levels only begins to appear in U1 cells.

The data for J1.1. are even less convincing.

Figure 2F. PMA is known to induce an oxidative stress response, however, in the experiments the data suggest that PMA results in a downregulated oxidative stress response. Maybe the authors could explain this discrepancy with the literature. In fact, both shRNA transductions, scr and CTH-specific seem to result in a lower PMA response.

Given that the others in subsequent experiments use GYY4137, which is supposed to mimic the increased release of H2S, the authors should have definitely included experiments in which they would overexpress CTH, e.g. by retroviral transduction. Specifically in U1 cells, which seemingly do not express CBS, overexpression of CBS should also result in a suppressed phenotype.

Figure 4F: The authors need to explain how they can measure a 4-fold gag RNA expression change in untreated cells. Also, according to Figure 4A, 300 µM GYY produces much less H2S than 5mM, yet the suppressive effect of 300 µM GYY is much higher?

Initially, the authors argue "that the depletion of CTH could contribute to redox imbalance and mitochondrial dysfunction to promote HIV-1 reactivation"(p. 9). Less CTH would suggest less produced H2S. However, later on in the manuscript they demonstrate that addition of a H2S source (GYY4137) results in the suppression of HIV-1 replication and supposedly HIV-1 reactivation. This is somewhat confusing.

CTH, or for that matter CBS or MPST do not only produce H2S, however, they also are part of other metabolic pathways. It would have been interesting and important to study how these metabolic pathways were affected by the genetic manipulations and also how the increased presence of H2S (GYY4137) would affect the metabolic activity of these enzymes or their expression.

H2S has been reported to cause NFkB inhibition by sulfhydration of p65; as such, the findings here are not particularly novel or surprising. Also, H2S induced sulfhydration is rather not targeted to a specific protein, let alone a HIV protein, making this approach a very unlikely alternative to current ART forms.

The description of the primary T cell model used to generate the data in Figure 6 is slightly misleading. Also, the idea of this model was originally to demonstrate that "block and lock" by didehydro-cortistatin is possible. In this application, the authors did not investigate whether GYY4137 would actually induce a HIV "block and lock" over an extended period of time.

The authors claim that the major conclusion of their study is that HIV-1 reactivation is coupled to depletion of endogenous H2S, which is associated with dysfunctional mitochondrial bioenergetics, in particular suppressed OXPHOS, GSH/GSSG imbalance, and elevated mitoROS. However, the authors never provide evidence that endogenous H2S is altered in latently HIV-1 infected cells (which may actually be an impossible task). By the end of the manuscript, the authors have not provided clear evidence that the effects of e.g. CTH deletion would be mediated by the production of H2S, and not by another function of the enzyme. Similarly, the inability of stimuli to trigger efficient HIV-1 reactivation following the provision of unnaturally high levels of H2S is not surprising given reports on the effect of GYY4137 as anti-inflammatory agent and suppressor NF-κB activation. Unless the authors were to demonstrate a true "block and lock" effect by GYY4137 the data will likely have limited impact on the HIV cure field.

[Editors’ note: further revisions were suggested prior to acceptance, as described below.]

Thank you for choosing to send your work entitled "Hydrogen sulfide blocks HIV rebound by maintaining mitochondrial bioenergetics and redox homeostasis" for consideration at *eLife*. Your letter of appeal has been considered by a Senior Editor and a Reviewing editor, and we are prepared to consider a revised submission with no guarantees of acceptance.

Upon careful consideration of your letter and after discussion with reviewers, we have the following recommendations.

We recognize that there are several strong aspects of the work presented in your manuscript however, the possible clinical implications of the work are overstated. In addition, further clarification is required in terms of the molecular mechanism proposed. We are willing to consider a revised manuscript based on the following provisors:

1. Regarding the conclusion that the reactivation of HIV-2 associated with CTH knockdown must be due to an absence of H2S: This conclusion is problematic as CTH has other functions as well. In addition, other proteins can produce H2S. The addition of H2S in the form of GYY4137, and associated inhibition of HIV-1 reactivation, does not bring further clarity as the concentrations of H2S used are much higher than anything the cells could produce by themselves. As CTH catalyses the last step in the trans-sulfuration pathway from methionine to cysteine, an important experiment would be to demonstrate that CHT KO is specific for H2S production and does not affect cysteine levels. Also demonstrating that cysteine deprivation does not lead to HIV-1 reactivation due to a generalised stress response would be important.

2. Cell cultures produce so little H2S that it cannot be measured. As such, in the experiments, addition of GYY4137 produce completely unphysiological H2S concentrations. Given that protein sulfurylation affects thousands of proteins, the addition of GYY4137 is not a biologically relevant reproduction of the presence of CTH, as the experimental system may be associated proteome-wide activity changes. H2S donors are mostly considered for cancer and anti-inflammatory treatments, because they seem to be immunosuppressive. To translate these findings towards clinical relevance for a cure of latent HIV-1 infection, you would have to show that lack of reactivation is not associated with a lack of T cell response. For example, you could demonstrate that T cells, after being cultured in GYY4137 containing medium, still respond to anti-CD3 stimulation with IL-2 secretion or CD25 upregulation. A better assay would be that the extended presence of GYY4137 does not abrogate the ability of T cells to proliferate upon antigen recognition (CMV pp65 should be a good antigen in combination with CFSE stains).

3. Clinical Relevance: Unless GYY4137 produces a "lock" state of latent HIV-1 infection events, or selectively kills latently HIV-1 infected T cells, there is no clinical relevance for a cure. No one will exchange ART for an immuno-suppressive experimental treatment, that already in vitro is not sufficiently potent to completely block reactivation. We suggest you remove any focus in the discussion on possible clinical implications and approach presenting this study from a fundamental research perspective as you do have some compelling findings.

---

## [Author Response]

[Editors’ note: The authors appealed the original decision. What follows is the authors’ response to the first round of review.]

Reviewer #1:[…] 1. Problems of dosing and side-effects of H2S donors should be critically discussed. The advantage of combining an H2S donor with ART did not become clear to me since the latter very reliably prevents viral replication.

We appreciate the comment. We appropriately discussed dosing and the side effects of H_2_S donors in the revised manuscript (Line no. 619, page no. 25 in the revised manuscript). The higher concentration of H_2_S is associated with toxicity and side effects (2), whereas lower micro molar concentrations are beneficial (3). The absorption and metabolism are reasonably well characterized for exogenously administered high doses of H_2_S (4,5). The sulfide oxidation pathways is involved in oxidation of excess H_2_S to thiosulphate and sulphate, which are cleared from the body via the urine in free or conjugated form (6). We assessed cytotoxicity of H_2_S donor (GYY4137) in all cell lines and primary CD4^+^ T cell model and selected non-toxic concentrations of GYY4137 to perform assays. As suggested by the editors, we have toned down the clinical potential of H_2_S donor in the revised manuscript.

2. The study is comprehensive and it is appreciated that the authors analyzed the impact of an H2S donor in primary CD4^+^ T cells. However, all mechanistic data were obtained in cell lines that may not faithfully represent the status of latently HIV infected cells in vivo. Ideally, some key finding should be confirmed in primary cells. Otherwise this limitation should be mentioned. Similarly, the cell line used harbor copies of T cell line adapted HIV-1 strains that may differ from proviruses representing the latent HIV reservoirs in vivo.

We appreciate the reviewer’s concern. For the reasons outlined by the reviewer, we examined the effect of GYY4137 on the CD4^+^ T cells derived from latent HIV-1 infected patients. Consistent with the cell line data, GYY4137 maintains mitochondrial bioenergetics (OXPHOS), suppresses mitochondrial reactive oxygen species (ROS), and induces the expression of the Nrf2-dependent antioxidant genes (Figure 6 and 7 in the revised manuscript). In the revised manuscript, we further extended these observations and showed that the effect of GYY4137 is not due to the impaired CD4^+^ T cells activation (Figure 6—figure supplement 2A-J in the revised manuscript).

To further allay the reviewer’s concern, we isolated primary CD4^+^ T cells from healthy donors and assessed the influence of GYY4137 on viral replication. Similar to findings in cell lines, GYY4137 diminished HIV-1 replication in primary CD4^+^ T cells without affecting cellular viability (Figure 4G and Figure 4—figure supplement 1E in the revised manuscript).

3. The authors propose to repurpose H2S donors for clinical treatment. Are such agents already used in the clinic or currently just examined in preclinical animal models?

Several H_2_S donors are in human clinical trials. A list of donors and the progress made in clinical trials are indicated in Author response table 1. We have included this information in the revised manuscript (Line no. 627, page no. 26 in the revised manuscript)

**Author response table 1. sa2table1:** List of H_2_S releasing donors and the progress made in clinical trials.

SI no.	Name of H2S donors	Company	Characteristic	Potential application	Development stage
1	ACS-15	CTG Pharma	Diclofenac derivative (NSAID)	Arthritis	Pre-clinical
2	ATB-346	Antibe	Naproxen derivative (NSAID)	Osteoarthritis	Phase 3
3	ATB-429	Antibe	NSAID-coupled	Inflammatory bowel disease	Pre-clinical
4	SG1002	Sulfagenix		Heart failure	Phase 2
5	IK-1001	Ikaria	Na2S	Multiple hypoxic/ischaemic condition	Phase 2
6	GYY4137		Lawesson’s reagent	Parkinson’s disease	Pre-clinical

4. The authors state that H2S "bolsters the anti-HIV potential of immune cells". This conclusion does not seem to be fully warranted based on the data presented. Thus, the statement should be cautioned or further substantiated.

We have rephrased the sentence to reflect our data showing the ability of GYY4137 to reduce the chances of viral rebound.

Reviewer #2:While the idea of H2S acting as a regulator of HIV-1 latency is interesting, there is a long list of inconsistencies that hamper the enthusiasm for this manuscript.Much of the cited literature that is used to make the case for their hypothesis is very old and actually refers to active HIV infection and patient studies prior to ART. Also, the literature they cite regarding the role of H2S as an antimicrobial agent seem to be limited to tuberculosis infection.

We have revised the list of literature and included more relevant references post-ART era. Recently, the antimicrobial role of H_2_S is comprehensively examined in the context of tuberculosis. Given the close association of TB with HIV, we thought our study is very timely and essential. However, we would like to point out that the references showing the effect of H_2_S on infection caused by respiratory viruses are included in the manuscript (7-9). Further, recent findings showing the influence of H_2_S in the context of SARS-CoV2 infection are also included in the revised manuscript (Line no. 592, page no. 24 in the revised manuscript).

The choice of the latently infected model cell lines is rather unfortunate. There are much better defined models out there these days than J1.1 or U1 cells, such as the J-LAT cells from the Verdin lab or the various reporter cell lines generated by Levy and co-workers. In particularly, U1 cells should not be considered as latently infected, as the virus has a defect in the Tat/TAR axis and is mostly just transcriptionally attenuated. It is unclear why the authors only use J-LAT cells for one of the last experiments.

As suggested by the reviewer, we have generated new data using J-LAT cells in the revised manuscript. First, we confirmed that PMA-mediated HIV-1 reactivation in J-LAT cells is associated with the down-regulation of *cbs*, *cth*, and *mpst* transcripts (Figure 1—figure supplement 1C-D in the revised manuscript). Additionally, we have performed several other mechanistic experiments in J-LAT cells to validate the data generated in U1 (see response below).

It is further unclear why the authors perform most of the experiments using U1 cells, which are considered promonocytic, but in the end seek to demonstrate the influence of H2S on latent HIV-1 infection in CD4 T cells. Performing all experiments in J1.1 or better J-LAT cells would have seemed more intuitive.

The choice of U1 was based on our earlier studies showing that U1 cells uniformly recapitulate the association of redox-based mechanisms and mitochondrial bioenergetics with HIV-latency and reactivation (10-12). We have validated key findings of U1 cells in J1.1 and J-Lat cell lines. We genetically and chemically silenced the expression of CTH in J-Lat cells and examined the effect on HIV-1 reactivation. Consistent with U1 and J1.1, genetic silencing of CTH using CTH-specific shRNA (shCTH) reactivated HIV-1 in J-Lat (Figure 2—figure supplement 1F-G in the revised manuscript). Supporting this, pre-treatment of J-Lat with non-toxic concentrations of a well-established CTH inhibitor, propargylglycine (PAG) further stimulated PMA-induced HIV-1 reactivation (Figure 2—figure supplement 1H-I in the revised manuscript). Altogether, using various cell line models of HIV-1 latency, we confirmed that endogenous H_2_S biogenesis counteracts HIV-1 reactivation.

The authors suggest that H2S production would control latent HIV-1 infection and reactivation. Regarding the idea that CBS, CTH or possibly MPST would control latent infection as a function of their ability to produce H2S from different sources, there are several questions. First, if H2S is the primary factor, why would the presence of e.g. MPST no compensate for the reduction of CTH? Second, why would J1.1 and U1 cells both host latent HIV-1 infection events, however, their CDB/CTH/MPST composition is completely different? Third, natural variations in CTH expression caused by culture over time are larger than variations caused by PMA activation.

These questions are important and complex. CBS, CTH, and MPST produce H_2_S in the sulfur network. CBS and CTH reside in the cytoplasm, whereas MPST is mainly involved in cysteine catabolism and is mitochondrial localized. The lack of compensation of CTH by MPST could be due to the compartmentalization of their activities. Furthermore, CTH and CBS activities are regulated by diverse metabolites, including heme, S-adenosyl methionine (SAM), and nitric oxide/carbon monoxide (NO/CO). In contrast, MPST activity responds to cysteine availability. How substrates/cofactors availability and enzyme choices are regulated in the cellular milieu of J1.1 and U1 is an interesting question for future experimentation.

Moreover, the tissue-specific expression/activity of CBS and CTH dictates their relative contributions in H_2_S biogenesis and cellular physiology (13). Some of these factors are likely responsible for differential expression of CBS, CTH, and MPST in J1.1 and U1 cells. Regardless of these concerns, viral reactivation uniformly reduces the expression of CTH in U1, J1.1, and J-Lat. While we cannot completely rule out natural variations in CTH expression over prolonged culturing, in our experimental setup CTH remained stably expressed and consistently showed down-regulation upon PMA treatment as compared to untreated conditions.

Also, the statement that H2S production as exerted per loss of CTH would control reactivation is not supported by the kinetic data. In latently HIV-1 infected T cell lines or monocytic cell lines, PMA-mediated HIV-1 reactivation at the protein level is usually almost complete after 24 hours, but at this time point the difference between e.g. CTH levels only begins to appear in U1 cells.The data for J1.1. are even less convincing.

We have performed the kinetics of p24 production and CTH in U1 cells. We showed that the levels of p24 gradually increased from 6 h and kept on increasing till the last time point, i.e., 36 h post-PMA-treatment (Figure 2D in the revised manuscript). The p24 ELISA detected a similar kinetics of p24 increase in the cell supernatant (Figure 2E in the revised manuscript). The CTH levels show reduction at 24 h and 36 h. Based on these data, we report that HIV-1 reactivation is associated with diminished biogenesis of endogenous H_2_S. We have not made any claims that depletion of CTH precedes HIV reactivation. However, our CTH knockdown data clearly showed that diminished expression of CTH reactivates HIV-1 in the absence of PMA, which is consistent with our hypothesis that H_2_S production is likely to be a critical host component for maintaining viral latency.

Figure 2F. PMA is known to induce an oxidative stress response, however, in the experiments the data suggest that PMA results in a downregulated oxidative stress response. Maybe the authors could explain this discrepancy with the literature. In fact, both shRNA transductions, scr and CTH-specific seem to result in a lower PMA response.

In our experiment, PMA treatment for 24 h results in down-regulation of oxidative stress genes. However, the effect of PMA on the oxidative stress responsive genes is time-dependent. In our earlier publication, we showed that 12 h PMA treatment induces oxidative stress responsive genes in U1 cells (12), whereas at 24 h, the expression of genes is down-regulated (10). Genetic silencing of CTH resulted in elevated mitochondrial ROS and GSH imbalance, which is in line with a further decrease in the expression of oxidative stress responsive genes as compared to PMA alone. As a consequence, PMA-treatment of U1-shCTH induced HIV-1 reactivation, which supersedes that stimulated by PMA or shCTH alone.

Given that the others in subsequent experiments use GYY4137, which is supposed to mimic the increased release of H2S, the authors should have definitely included experiments in which they would overexpress CTH, e.g. by retroviral transduction. Specifically in U1 cells, which seemingly do not express CBS, overexpression of CBS should also result in a suppressed phenotype.

We have explored the role of elevated H_2_S levels using GY44137. Treatment with GYY4137 suppressed HIV reactivation in multiple cell lines and primary CD4^+^ T cells. As suggested by the reviewer, overexpression of CTH could be another strategy to validate these findings. However, since the transsulfuration pathway and active methyl cycle are interconnected and share metabolic intermediates (e.g., homocysteine), overexpression of CTH could disturb this balance and may lead to metabolic paralysis. Owing to these potential limitations, we used a slow releasing H_2_S donor (GYY4137) to chemically complement CTH deficiency during HIV reactivation. We thank the reviewer for this comment.

Figure 4F: The authors need to explain how they can measure a 4-fold gag RNA expression change in untreated cells. Also, according to Figure 4A, 300 µM GYY produces much less H2S than 5mM, yet the suppressive effect of 300 µM GYY is much higher?

The four-fold-expression in untreated cells is likely due to leaky control of viral transcription in J1.1 cells (14 16). However, to avoid confusion, we have replotted the results by normalizing the data generated upon PMA mediated HIV reactivation with the PMA untreated cells in the revised manuscript (Figure 4F in the revised manuscript). The suppressive effect of GYY4137 at the lower concentration is intriguing but consistent with the findings that high and low concentrations of H_2_S have profound and distinct effects on cellular physiology (3,17). One possibility is that the high concentration of H_2_S induces mitochondrial sulfide oxidation pathway to avert toxicity. This might modulate mitochondrial activity and ROS, resulting in the suppression of GYY4137 effect. Consistent with this, higher concentrations of H_2_S have been shown to cause pro-oxidant effects, DNA damage and genotoxicity (3,18). We have discussed these possibilities in the revised manuscript (Line no. 329, page no. 14 in the revised manuscript)

Initially, the authors argue "that the depletion of CTH could contribute to redox imbalance and mitochondrial dysfunction to promote HIV-1 reactivation"(p. 9). Less CTH would suggest less produced H2S. However, later on in the manuscript they demonstrate that addition of a H2S source (GYY4137) results in the suppression of HIV-1 replication and supposedly HIV-1 reactivation. This is somewhat confusing.

We show that depletion of endogenous H_2_S by diminished expression of CTH (U1-shCTH) resulted in higher mitochondrial ROS and GSH/GSSG imbalance. Both of these alterations are known to reactivate HIV-1 and promote replication (10,11,19). The addition of GYY4137 chemically compensated for the diminished expression of CTH, and prevented HIV-1 reactivation in U1-shCTH. These events are expected to suppress HIV-1 replication and reactivation. We have made this distinction clear in the revised manuscript (line no. 284, page no. 12 in the revised manuscript).

CTH, or for that matter CBS or MPST do not only produce H2S, however, they also are part of other metabolic pathways. It would have been interesting and important to study how these metabolic pathways were affected by the genetic manipulations and also how the increased presence of H2S (GYY4137) would affect the metabolic activity of these enzymes or their expression.

We fully agree with the reviewer. In fact, our NanoString data show that upon CTH knockdown (U1-shCTH), MPST levels were down-regulated and CBS remained undetectable (Figure 2F in the revised manuscript). Additionally, GYY4137 treatment induced the expression of CTH but not MPST upon PMA addition (Figure 5A in the revised manuscript). We have incorporated these findings in the revised manuscript (line no. 215, page no. 9 and line no. 375, page no. 15 in the revised manuscript). Given that CBS and CTH catalyzed at least eight H_2_S generating steps and two cysteine-producing reactions, the modulation of CTH by HIV is likely to have a widespread influence on transsulfuration pathway and active methyl cycle intermediates. Our future strategies are to generate a comprehensive understanding of sulfur metabolism underlying HIV latency and reactivation. These experiments require multiple biochemical and genetic technologies with appropriate controls. We hope that the reviewer would agree with our views that these experiments should be a part of future investigation. We thank the reviewer for this comment.

H2S has been reported to cause NFkB inhibition by sulfhydration of p65; as such, the findings here are not particularly novel or surprising. Also, H2S induced sulfhydration is rather not targeted to a specific protein, let alone a HIV protein, making this approach a very unlikely alternative to current ART forms.

We believe that NF-κB inhibition is not the only mechanism by which H_2_S exerts its influence on HIV latency. Recent studies point towards the importance of the Nrf2-Keap1 axis in sustaining HIV-latency (20). Our data suggest an important role for Nrf2-Keap1 signaling in mediating the influence of H_2_S on HIV latency. Additionally, recruitment of an epigenetic silencer YY1 is also affected by H_2_S. Interestingly, YY1 activity is modulated by redox signaling (21), suggesting H_2_S could be an important regulator of YY1 activity in HIV-infected cells. We have so far, no evidence for viral proteins targeted by H_2_S. However, experiments to examine global S-persulfidation of host and HIV protein are ongoing in the laboratory to fill this knowledge gap. Lastly, our findings raise the possibility of exploring H_2_S donors with the current ART (not as an alternate to ART) for reducing virus reactivation. We have tone down the clinical relevance of our findings.

The description of the primary T cell model used to generate the data in Figure 6 is slightly misleading. Also, the idea of this model was originally to demonstrate that "block and lock" by didehydro-cortistatin is possible. In this application, the authors did not investigate whether GYY4137 would actually induce a HIV "block and lock" over an extended period of time.

As suggested by the reviewer, we have cited the didehydro-cortistatin studies as the basis of our strategy. Our idea was to adapt the primary T cell model to begin understanding the role of H_2_S in blocking HIV rebound. Our results indicate the future possibility of investigating GYY4137 to lock HIV in deep latency for an extended period of time. However, comprehensive investigation would require long-term experiments and samples from multiple HIV subjects. In the current pandemic times with overburdened Indian clinical settings, we cannot plan these experiments. However, we hope our data form a solid foundation for HIV researchers to perform extended “block and lock” studies using H_2_S donors.

The authors claim that the major conclusion of their study is that HIV-1 reactivation is coupled to depletion of endogenous H2S, which is associated with dysfunctional mitochondrial bioenergetics, in particular suppressed OXPHOS, GSH/GSSG imbalance, and elevated mitoROS. However, the authors never provide evidence that endogenous H2S is altered in latently HIV-1 infected cells (which may actually be an impossible task). By the end of the manuscript, the authors have not provided clear evidence that the effects of e.g. CTH deletion would be mediated by the production of H2S, and not by another function of the enzyme. Similarly, the inability of stimuli to trigger efficient HIV-1 reactivation following the provision of unnaturally high levels of H2S is not surprising given reports on the effect of GYY4137 as anti-inflammatory agent and suppressor NF-κB activation. Unless the authors were to demonstrate a true "block and lock" effect by GYY4137 the data will likely have limited impact on the HIV cure field.

It's difficult to measure H_2_S levels in the latently infected primary cells due to the assay's sensitivity and the insufficient number of cells latently infected with HIV-1. However, in the revised manuscript we have clearly shown that cysteine levels are not affected by CTH depletion and cysteine deprivation does not reactivate HIV-1. These results indicate that the effects of CTH depletion are likely mediated by H_2_S. This is consistent with our data showing that GYY4137 specifically complement CTH deficiency and blocks HIV-1 reactivation in U1-shCTH. Further, we carried in-depth investigation to show that the effect of GYY4137 is not due to impaired activation of CD4^+^ T cells.

Lastly, since CTH catalyzed multiple reactions during H_2_S production, we cannot rule out the effect of other metabolites in this process. However, we think that this is outside the scope of the present study. Our study focuses on understanding of how H_2_S modulates redox, mitochondrial bioenergetics, and gene expression in the context of HIV latency. These understandings are likely to positively impact future studies exploring the role of H_2_S on HIV cure.

References:

1. Zhu, J., Berisa, M., Schwörer, S., Qin, W., Cross, J. R., and Thompson, C. B. (2019) Transsulfuration Activity Can Support Cell Growth upon Extracellular Cysteine Limitation. Cell Metabolism 30, 865-876.e865

2. Shen, Y., Shen, Z., Luo, S., Guo, W., and Zhu, Y. Z. (2015) The Cardioprotective Effects of Hydrogen Sulfide in Heart Diseases: From Molecular Mechanisms to Therapeutic Potential. Oxid Med Cell Longev 2015, 925167

3. Szabo, C., Ransy, C., Módis, K., Andriamihaja, M., Murghes, B., Coletta, C., Olah, G., Yanagi, K., and Bouillaud, F. (2014) Regulation of mitochondrial bioenergetic function by hydrogen sulfide. Part I. Biochemical and physiological mechanisms. Br J Pharmacol 171, 2099-2122

4. R J Reiffenstein, W C Hulbert, a., and Roth, S. H. (1992) Toxicology of Hydrogen Sulfide. Annual Review of Pharmacology and Toxicology 32, 109-134

5. Beauchamp, R. O., Jr., Bus, J. S., Popp, J. A., Boreiko, C. J., and Andjelkovich, D. A. (1984) A critical review of the literature on hydrogen sulfide toxicity. Crit Rev Toxicol 13, 25-97

6. Kage, S., Takekawa, K., Kurosaki, K., Imamura, T., and Kudo, K. (1997) The usefulness of thiosulfate as an indicator of hydrogen sulfide poisoning: three cases. Int J Legal Med 110, 220-222

7. Li, H., Ma, Y., Escaffre, O., Ivanciuc, T., Komaravelli, N., Kelley, J. P., Coletta, C., Szabo, C., Rockx, B., Garofalo, R. P., and Casola, A. (2015) Role of hydrogen sulfide in paramyxovirus infections. J Virol 89, 5557-5568

8. Ivanciuc, T., Sbrana, E., Ansar, M., Bazhanov, N., Szabo, C., Casola, A., and Garofalo, R. P. (2016) Hydrogen Sulfide Is an Antiviral and Antiinflammatory Endogenous Gasotransmitter in the Airways. Role in Respiratory Syncytial Virus Infection. Am J Respir Cell Mol Biol 55, 684-696

9. Bazhanov, N., Escaffre, O., Freiberg, A. N., Garofalo, R. P., and Casola, A. (2017) Broad-Range Antiviral Activity of Hydrogen Sulfide Against Highly Pathogenic RNA Viruses. Scientific Reports 7, 41029

10. Bhaskar, A., Munshi, M., Khan, S. Z., Fatima, S., Arya, R., Jameel, S., and Singh, A. (2015) Measuring glutathione redox potential of HIV-1-infected macrophages. J Biol Chem 290, 1020-1038

11. Tyagi, P., Pal, V. K., Agrawal, R., Singh, S., Srinivasan, S., and Singh, A. (2020) Mycobacterium tuberculosis Reactivates HIV-1 via Exosome-Mediated Resetting of Cellular Redox Potential and Bioenergetics. mBio 11

12. Singh, S., Ghosh, S., Pal, V. K., Munshi, M., Shekar, P., Narasimha Murthy, D. T., Mugesh, G., and Singh, A. (2021) Antioxidant nanozyme counteracts HIV-1 by modulating intracellular redox potential. EMBO Mol Med 13, e13314

13. Liu, Y. H., Lu, M., Hu, L. F., Wong, P. T., Webb, G. D., and Bian, J. S. (2012) Hydrogen sulfide in the mammalian cardiovascular system. Antioxid Redox Signal 17, 141-185

14. Krishnan, V., and Zeichner, S. L. (2004) Host Cell Gene Expression during Human Immunodeficiency Virus Type 1 Latency and Reactivation and Effects of Targeting Genes That Are Differentially Expressed in Viral Latency. Journal of Virology 78, 9458-9473

15. Symons, J., Chopra, A., Malatinkova, E., De Spiegelaere, W., Leary, S., Cooper, D., Abana, C. O., Rhodes, A., Rezaei, S. D., Vandekerckhove, L., Mallal, S., Lewin, S. R., and Cameron, P. U. (2017) HIV integration sites in latently infected cell lines: evidence of ongoing replication. Retrovirology 14, 2

16. Butera, S. T., Roberts, B. D., Lam, L., Hodge, T., and Folks, T. M. (1994) Human immunodeficiency virus type 1 RNA expression by four chronically infected cell lines indicates multiple mechanisms of latency. J Virol 68, 2726-2730

17. Whiteman, M., Li, L., Rose, P., Tan, C. H., Parkinson, D. B., and Moore, P. K. (2010) The effect of hydrogen sulfide donors on lipopolysaccharide-induced formation of inflammatory mediators in macrophages. Antioxid Redox Signal 12, 1147-1154

18. Olas, B., Brodek, P., and Kontek, B. (2019) The Effect of Hydrogen Sulfide on Different Parameters of Human Plasma in the Presence or Absence of Exogenous Reactive Oxygen Species. Antioxidants (Basel) 8

19. El-Amine, R., Germini, D., Zakharova, V. V., Tsfasman, T., Sheval, E. V., Louzada, R. A. N., Dupuy, C., Bilhou-Nabera, C., Hamade, A., Najjar, F., Oksenhendler, E., Lipinski, M., Chernyak, B. V., and Vassetzky, Y. S. (2018) HIV-1 Tat protein induces DNA damage in human peripheral blood B-lymphocytes via mitochondrial ROS production. Redox Biol 15, 97-108

20. Shytaj, I. L., Lucic, B., Forcato, M., Penzo, C., Billingsley, J., Laketa, V., Bosinger, S., Stanic, M., Gregoretti, F., Antonelli, L., Oliva, G., Frese, C. K., Trifunovic, A., Galy, B., Eibl, C., Silvestri, G., Bicciato, S., Savarino, A., and Lusic, M. (2020) Alterations of redox and iron metabolism accompany the development of HIV latency. EMBO J 39, e102209

21. Bonavida, B., and Baritaki, S. (2012) Inhibition of Epithelial-to-Mesenchymal Transition (EMT) in Cancer by Nitric Oxide: Pivotal Roles of Nitrosylation of NF-kappaB, YY1 and Snail. For Immunopathol Dis Therap 3, 125-133

[Editors’ note: what follows is the authors’ response to the second round of review.]

Upon careful consideration of your letter and after discussion with reviewers, we have the following recommendations.We recognize that there are several strong aspects of the work presented in your manuscript however, the possible clinical implications of the work are overstated. In addition, further clarification is required in terms of the molecular mechanism proposed. We are willing to consider a revised manuscript based on the following provisors:1. Regarding the conclusion that the reactivation of HIV-2 associated with CTH knockdown must be due to an absence of H2S: This conclusion is problematic as CTH has other functions as well. In addition, other proteins can produce H2S. The addition of H2S in the form of GYY4137, and associated inhibition of HIV-1 reactivation, does not bring further clarity as the concentrations of H2S used are much higher than anything the cells could produce by themselves. As CTH catalyses the last step in the trans-sulfuration pathway from methionine to cysteine, an important experiment would be to demonstrate that CHT KO is specific for H2S production and does not affect cysteine levels. Also demonstrating that cysteine deprivation does not lead to HIV-1 reactivation due to a generalised stress response would be important.

We thank the editors for raising these concerns. To address these comments, we measured cysteine levels in cell lysates of CTH knock down U1 cells (U1-shCTH) and compared it with the control cells (U1-shNT). The cysteine levels remain comparable in the lysates derived from both the cell lines, suggesting that CTH depletion does not affect cysteine levels (Figure 2—figure supplement 1C in the revised manuscript). The data also indicate that other routes for cysteine biosynthesis (e.g., de novo and assimilatory pathways) likely compensate for CTH depletion in U1 cells. As suggested by the reviewer, we next assessed if cysteine deprivation reactivates HIV. To deprive cells for cysteine, we used RPMI medium lacking cysteine and supplemented with dialyzed FBS (Gibcoä Cat# 26400044, see Materials and methods) as per the published protocol (1). The U1 cells were cultured in complete RPMI medium or cysteine-free RPMI medium and *gag* transcript was measured at 6 h, 12 h, and 24 h. The expression of *gag* transcript was not significantly affected upon cysteine depletion, suggesting that HIV reactivation is not stimulated by deprivation of cysteine (Figure 2—figure supplement 1D in the revised manuscript).

2. Cell cultures produce so little H2S that it cannot be measured. As such, in the experiments, addition of GYY4137 produce completely unphysiological H2S concentrations. Given that protein sulfurylation affects thousands of proteins, the addition of GYY4137 is not a biologically relevant reproduction of the presence of CTH, as the experimental system may be associated proteome-wide activity changes. H2S donors are mostly considered for cancer and anti-inflammatory treatments, because they seem to be immunosuppressive. To translate these findings towards clinical relevance for a cure of latent HIV-1 infection, you would have to show that lack of reactivation is not associated with a lack of T cell response. For example, you could demonstrate that T cells, after being cultured in GYY4137 containing medium, still respond to anti-CD3 stimulation with IL-2 secretion or CD25 upregulation. A better assay would be that the extended presence of GYY4137 does not abrogate the ability of T cells to proliferate upon antigen recognition (CMV pp65 should be a good antigen in combination with CFSE stains).

This is an excellent recommendation. Accordingly, we cultured PHA/IL-2 expanded CD4^+^ T cell lines for 14 days in presence of ART or ART in combination with 100 μM GYY4137. Cells were then labelled with 0.5 μM cell trace violet (CTV) and stimulated with either CMV, anti-CD3/anti-CD28 beads (at a bead: cell ratio of 1:0.25) or Staphylococcal enterotoxin B (SEB;100 ng/ml). After three days, IL-2 (20 U/ml) supplemented medium was added to the cells and cultured for another two days. Proliferation was measured by CTV dilution and CD25 expression using flow cytometry after 5 days of culture (Figure 6—figure supplement 2A-D in the revised manuscript). We included activation with SEB, as it is a known superantigen that can trigger polyclonal T cell activation by binding to the T cell receptor at the variable region of the β-chain (relevant to our system). Hence, in these long-term polyclonal cultures where frequencies of T cells specific to any given antigen e.g., CMV may hugely vary between samples or CMV-specific cells may not be detected as shown by our data, we chose to more directly test CD4^+^ T cell survival and proliferative capacity using SEB and CD3/28 beads as stimuli. The data demonstrate that long-term culture of cells in the presence of GYY4137 does not reduce or interfere with the capacity of the cells to respond to activation by CD3/28 or in response to SEB (Figure 6—figure supplement 2A-D in the revised manuscript).

To extend the suggestion of the reviewer further, we conducted another series of experiments to test the impact of GYY4137 in a more physiologically relevant short term T cell activation assay. Here PBMCs from HIV+ ART-treated patients were incubated with ART or ART in combination with 100 μM GYY4137 for 18 hr. Cells were analyzed for surface expression of CD25 and CD69 and intracellular staining of IFNγ by flow cytometry. The comprehensive gating strategy and representative flow cytometry plots are depicted in Figure 6—figure supplement 3A-3B of the revised manuscript. The data generated demonstrate no difference between GYY4137-treated versus untreated cultures in terms of CD4^+^ T cell or CD8^+^ T cell response to CMV peptide stimulation as measured by IFNγ expression or in the capacity of CD4^+^ and CD8^+^ T cells to express CD25 or CD69 in response to either CD3/28 or CMV stimulation (Figure 6—figure supplement 2E-J in the revised manuscript). Collectively, these findings demonstrate that CD4^+^ T cell function in both short-term and long-term cultures is preserved in the presence of GYY4137.

3. Clinical Relevance: Unless GYY4137 produces a "lock" state of latent HIV-1 infection events, or selectively kills latently HIV-1 infected T cells, there is no clinical relevance for a cure. No one will exchange ART for an immuno-suppressive experimental treatment, that already in vitro is not sufficiently potent to completely block reactivation. We suggest you remove any focus in the discussion on possible clinical implications and approach presenting this study from a fundamental research perspective as you do have some compelling findings.

We agree with the reviewer. We have removed the statement indicating possible clinical implications of these findings.